# PoPI: A machine learning-based consensus mechanism for blockchain-enabled IoT systems

**Mubtasim Kamal Dihan**, **Abdullah**, **Amina, Faisal Hussain, Md Moniruzzaman**, **Md Sakhawat Hossen***

Department of Computer Science and Engineering, Islamic University of Technology, Gazipur, Bangladesh

* sakhawat@iut-dhaka.edu

## Abstract

Internet of Things (IoT) enables seamless connectivity and intelligent automation across diverse domains, from healthcare and agriculture to smart cities and industrial systems. However, conventional IoT architectures often rely on centralized servers for data processing and coordination, resulting in poor scalability and decreased system reliability. The integration of blockchain with IoT offers a promising approach to address these limitations of centralized IoT architectures. In practice, however, existing blockchain consensus mechanisms are often unsuitable for resource-constrained IoT devices and dynamic network conditions. To overcome this limitation, we propose Proof of Periodic Inference (PoPI), a machine learning model-based consensus mechanism tailored for blockchain-based IoT systems. PoPI uses a supervised machine learning model to periodically select a group of block producers and maintains security through random block generation within the group. It incorporates both static and dynamically changing device features, such as battery level and resource usage, to select capable nodes with high reliability, and implements fair participation mechanisms to balance network involvement over time. Theoretical analysis and experimental evaluation demonstrate that PoPI achieves high scalability, low latency and improved applicability compared to the state-of-the-art consensus protocols in dynamic IoT environments.

## Introduction

For centuries, technological advancements have been motivated by the goal of developing intelligent systems that make human activities more efficient and improve quality of life [1]. Over the past few decades, the Internet of Things (IoT) has revolutionized how digital systems interact with the physical world and has gradually become an integral part of modern lifestyles and technological infrastructures [2]. However, most existing IoT architectures still depend on centralized cloud servers for data storage and processing. This centralized dependency introduces critical limitations, including a single point of failure, reduced scalability, and high operational

**Data availability statement:** All relevant data are contained within the manuscript.

**Funding:** The author(s) received no specific funding for this work.

**Competing interests:** The authors have declared that no competing interests exist.

costs as the number of connected devices increases [3]. Therefore, addressing these challenges has become essential for ensuring the long-term resilience and efficiency of IoT ecosystems.

Originally introduced as the foundational technology behind the Bitcoin cryptocurrency, blockchain has emerged as a powerful solution that can address several challenges faced by traditional IoT systems [4]. As a decentralized, distributed ledger technology, blockchain allows transactions to be recorded securely, transparently, and in a tamper proof manner across a network of nodes. Although blockchain technology was initially applied to cryptocurrencies, its core properties, including decentralization, immutability, and trustless consensus, enable its use in diverse domains such as smart cities, healthcare, and agriculture [5]. When integrated with IoT, blockchain can effectively mitigate the issues of centralization, enhance data integrity, and improve trust and scalability within interconnected device networks [6].

Consensus mechanisms lie at the heart of blockchain technology. They enable blockchain to provide security and trust equivalent to centralized systems without relying on a single authority [7]. By requiring the agreement of the majority of network participants on the validity and compliance of transactions or data, these mechanisms ensure a consistent and reliable ledger across all nodes [8]. However, most existing consensus mechanisms are not designed for the resource constraints of IoT devices [9]. Moreover, they generally overlook the continuously changing and uncertain conditions that affect the reliability of IoT devices in participating in consensus [10], leaving a gap in mechanisms suitable for such scenarios.

Battery life is one of the most critical concerns in IoT devices. For example, devices in environmental monitoring may be placed in remote areas with no access to power, making frequent recharging or replacement unfeasible and leading to premature battery failure or unexpected shutdowns [11]. Even devices with high capacity and recharge rates can drain to critical levels due to temporary issues like power outages or extended cloudy periods [12]. Beyond battery level, residual energy, charging rate, duty-cycle behavior, and thermal conditions can affect node availability. Network factors, such as reduced bandwidth or intermittent connectivity, can make devices temporarily unreachable. In addition, devices may enter low-power states to save energy or prioritize other tasks. Therefore, it is important to consider both energy status and a device's ability to participate at a given moment. Given these requirements, a consensus mechanism tailored for Blockchain-enabled IoT systems must overcome poor scalability, low efficiency, and the challenges posed by the dynamic operating conditions of IoT devices [13].

To address these requirements, we propose a machine learning-based consensus mechanism called Proof of Periodic Inference (PoPI) for Blockchain-enabled IoT systems. Specifically, we make the following three contributions in this work.

1. We minimize the computational overhead of repeatedly selecting a single block producer by selecting a group of block producers instead, each responsible for generating one block during a defined period. Security is ensured through a

randomized block production order within the group. The size of the group scales proportionally to the total number of nodes.

2. We incorporate the dynamically changing status of IoT devices into the block producer selection process, ensuring that only nodes with sufficient current capability participate, thereby reducing overall latency. Furthermore, we design a mechanism for efficient and reliable sharing of this dynamic information across the network.

3. We implement fair participation strategies to prevent recently selected nodes from being repeatedly chosen, while allowing new nodes to join the consensus process after a suitable interval.

By combining these strategies, PoPI achieves high scalability, broad applicability, and strong resilience in dynamic IoT ecosystems. The effectiveness of PoPI is justified through theoretical analysis and confirmed by experimental evaluation, which involves comparison with five other state-of-the-art consensus mechanisms using a discrete-event simulator.

The remainder of this paper is organized as follows. In Related Works section, we provide a literature review and identify research gaps in existing consensus mechanisms to justify our motivation. The Proposed Consensus Mechanism section describes PoPI in detail. In Theoretical Analysis section, we theoretically analyze the properties of PoPI. The experimental setup used to evaluate its performance is detailed in Experimental Design section. The Results & Discussion section presents the results of the performance evaluation. Finally, the Conclusion section offers concluding remarks, discusses limitations, and suggests directions for future research.

## Related Works

This section provides a brief overview of related works, organized into four categories. First, we explore the potential of blockchain beyond cryptocurrency, followed by a discussion of efforts to integrate machine learning into blockchain systems. Next, we examine traditional consensus frameworks and their limitations in IoT environments. Finally, we review modern consensus mechanisms that enhance traditional approaches for specific use cases.

### Blockchain beyond cryptocurrency

Although blockchain technology emerged through its use in cryptocurrency, its significant growth in academic interest in recent years can be attributed to its immense potential to improve a wide range of traditional systems beyond the cryptocurrency domain [14]. In healthcare, blockchain has been explored to enhance the interoperability of medical information exchange between organizations as well as strengthen the security of stored data [15,16]. Xiong *et al.* [17] have discussed how blockchain can improve various aspects of agriculture, including smart farming, food supply chains, and agricultural insurance. Haque *et al.* [18] have proposed a blockchain-based alternative to GitHub for efficient source code storage. Xu *et al.* [19] have provided valuable insights into decentralized finance using a hybrid bibliometric and TCM framework. In the accounting domain, blockchain-based frameworks have been introduced to strengthen the intelligence, efficiency, and quality of internal auditing processes [20]. Additionally, several studies have demonstrated how IoT can leverage blockchain to transform diverse systems [21,22]. Overall, these works highlight the effectiveness of blockchain across various non-cryptocurrency applications and its strong potential in IoT-enabled environments.

### Machine learning in blockchain

In the current century, machine learning (ML) has become increasingly integrated in nearly every technological domain [23]. Following this trend, researchers have explored how ML can complement blockchain to address limitations in scalability, security, and efficiency [24]. Different sectors such as supply chain management [25–27], healthcare [28,29], agriculture [30], education [31] and even sports [32] have explored the integration of ML and blockchain, often in combination with IoT, to improve transparency, automation, and data reliability. In finance, the advanced information processing

 

capabilities of Large Language Models (LLMs) have been leveraged to enhance blockchain-driven supply chain financing platforms [33]. ML techniques originally developed for anomaly detection in conventional systems have also been applied to blockchain to detect irregular or malicious behaviors that could compromise network security [34,35]. Beyond domain-specific applications, ML has been utilized to optimize fundamental blockchain components. For example, Chesuh *et al.* [36] addressed Bitcoin's scalability limitations by employing Extreme Gradient Boosting (XGB) to dynamically predict optimal block sizes. These studies demonstrate that the combinations of blockchain and ML can be very powerful and, with further research and development, can transform numerous real-world domains.

### Traditional consensus mechanisms

Traditionally, most of the early blockchain consensus mechanisms were developed primarily for cryptocurrency systems. Proof of Work (PoW) [37], introduced as the consensus protocol for the Bitcoin network, remains the most well-known mechanism in blockchain technology. Although PoW offers a decentralized and secure framework, it demands substantial computational resources as its mining operation requires solving complex cryptographic puzzles to achieve a specific hash difficulty. Mechanisms such as Proof of Stake (PoS) [38] and Delegated Proof of Stake (DPoS) [39] rely on stake-based validation, where nodes are selected to validate and produce blocks based on the amount of cryptocurrency they hold or are delegated by other participants. Althouggh these approaches significantly reduce the computational overhead compared to PoW, stake-based selection does not necessarily reflect the reliability or security requirements of IoT networks. Practical Byzantine Fault Tolerance (PBFT) [40] is a consensus mechanism designed for blockchain networks that allows nodes to reach agreement even in the presence of Byzantine (malicious or faulty) participants through multiple rounds of message exchange among validators. However, the communication overhead of these approaches increases quadratically with the number of nodes, making it unsuitable for IoT environments where devices may generate hundreds of transactions per minute. Other traditional consensus protocols, including Hyperledger Fabric [41], RAFT [42], and Ripple [43], are designed primarily for permissioned blockchains, where node participation is restricted. In summary, most traditional consensus algorithms fail to meet the scalability, efficiency, and adaptability requirements of IoT ecosystems.

### Modern consensus mechanisms

To meet the evolving requirements of decentralized systems and address the shortcomings of traditional consensus mechanisms, several innovative approaches have been proposed in recent years. For instance, Dynamic Trust DPoS (DT-DPoS) [44] incorporates an EigenTrust-based model to represent stake sharing and stake values as measures of mutual trust among nodes. Other trust-oriented mechanisms, such as Proof of Trust (PoT) [45] and Proof of Reputation (PoR) [46], similarly rely on trust scores to evaluate node behavior. While these approaches provide a partial indication of node reliability, trust alone is insufficient to capture the full dynamics and uncertainties of IoT environments. Proof of Elapsed Time (PoET) [47] mitigates computational overhead by using randomized wait times, where each participating node is assigned a randomly generated delay and the node with the shortest waiting time gains the right to produce the next block. However, as the number of nodes grows, wait times increase to prevent block collisions, reducing scalability in high-transaction IoT ecosystems.

Newer variants of PoW based on resource or computation have attempted to address its inefficiency by utilizing alternative forms of measurable resources. Mechanisms such as Proof of Useful Work (PoUW) [48], which requires nodes to perform meaningful computational tasks, Proof of Quality of Service (PoQ) [49], which evaluates nodes based on their service performance, and Proof of Space (PoSpace) [50], which relies on storage capacity instead of computational power, aim to improve resource efficiency of PoW. However, these mechanisms still face limitations when applied to IoT systems, as such devices are inherently resource-constrained. Several modern PBFT variants [51] aim to reduce message complexity, but the consensus time still increases sharply with growing node count. PPoR [52] combines PBFT and PoR for V2V energy sharing, but its application is limited to the Internet of Electric Vehicles. CE-PBFT [53] incorporates

 

decision tree algorithms to classify nodes based on their credibility and assign appropriate responsibilities to each class. This method more effectively assesses node reliability but does not account for the challenging operating conditions of IoT devices.

The integration of machine learning into consensus mechanisms remains a relatively unexplored area. Mechanisms such as Proof of Learning (PoLe) [54] and Proof of Federated Learning (PoFL) [55] rely on nodes' ability to perform model training. Although these approaches can be energy-efficient, a device's capacity to train a model alone does not fully capture its reliability. Proof of Deep Learning (PoDL) [56] is designed to recycle energy in PoW by having nodes perform deep learning tasks, but it was developed for cryptocurrencies. Goh *et al.* [57] proposed TDCB-D3P, a deep reinforcement learning (DRL)-based consensus mechanism designed to ensure high throughput while maintaining a strong level of security in blockchain-enabled IoT networks. However, the computational complexity and resource requirements of DRL models make them unsuitable for most resource-constrained IoT devices. Proof of Evolutionary Model (PoEM) [58] uses a supervised learning model to rank nodes and select the top-ranked node as block producer. However, it requires full inference over all nodes for each selection and relies on fixed parameters, making it less suitable for dynamic IoT environments with continuously changing device states. Overall, these limitations of existing approaches highlight the need for a lightweight and adaptive consensus mechanism capable of evaluating node reliability under dynamic conditions.

## Proposed consensus mechanism

In this section, we provide a detailed description of how the proposed consensus mechanism, Proof of Periodic Inference (PoPI), operates. First, we present an overview of the workflow of PoPI. We then describe its three main contributions, followed by an explanation of how the machine learning model can be formulated as a consensus problem.

### PoPI workflow overview

Fig 1 illustrates the overall workflow of our proposed consensus mechanism, PoPI. The foundation of PoPI is a supervised machine learning model that predicts and ranks the suitability of each node aiming to become a block producer in the next cycle. Active nodes transmit their current hardware and network status to a small rotating set of nodes called supervisors.

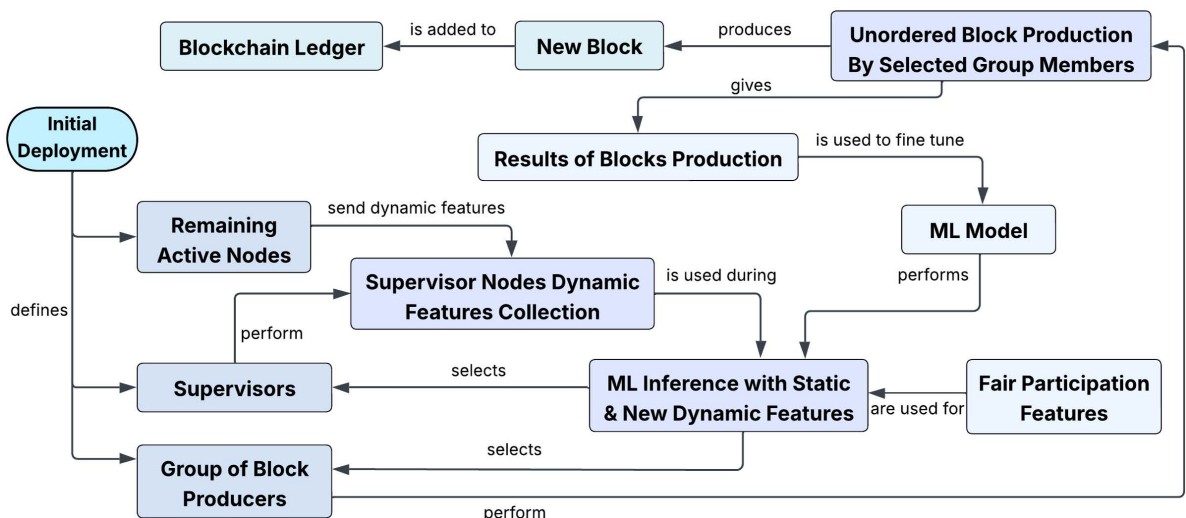

**Fig 1. High-level illustration of the workflow of our proposed mechanism.** During initial deployment, if no pretrained model is available, alternative methods assign roles to nodes. Later, supervisors apply ML models using different features to select the next supervisors and a group of block producers. Blocks are produced in a randomized order and the block production outcome is used to improve the ML model.

These data include metrics such as battery level, network bandwidth, and other operational parameters that assess each node's capability to successfully produce a block. As these parameters are highly dynamic, aggregating the current status across all nodes is necessary. The supervisor nodes perform this task efficiently while keeping communication overhead low.

Once supervisors receive status data from all candidate nodes, they use the latest machine learning model to predict the suitability score of each node. The resulting scores are broadcast to the network, and the *top-n* ranked nodes are selected as block producers for the upcoming cycle. Supervisors also package the inputs used for machine learning inference as transactions recorded on the blockchain, allowing participants to verify execution on authentic data. Multiple supervisors perform this task redundantly to prevent a single compromised supervisor from impairing network performance. The number of supervisors remains small, ensuring that system efficiency is not significantly affected. During this inference cycle, the model also identifies the next set of $s$ supervisors, distinct from the block producers, selected from nodes with the highest suitability scores that have successfully served as block producers at least once.

After the block producer list is finalized, block production begins. The highest-ranked node does not automatically assume the first position, as revealing the exact order could introduce security risks. Although all selected block producers participate in the cycle, their production order is determined through a secure randomization process. To avoid additional complexity in generating a globally consistent random value, this number is derived from the hash of the previous block, which is accessible to all nodes. The hash is converted into an integer, $r$, ranging from 1 to $n$, where $n$ represents the size of the block producer group. The $r^{th}$ node that has not yet produced a block is selected as the next block producer. Since all nodes possess the preceding block hash, they independently generate the same random value, ensuring consistent selection. Each subsequent producer learns its identity only after the previous block is completed, as the random value cannot be computed beforehand without the hash of a completed block, reinforcing system robustness.

By the time the current group of block producers completes block generation, the newly selected supervisors begin collecting requests from active nodes wishing to participate as block producers in the next cycle. Although all nodes in an IoT system should ideally participate in consensus, some may remain inactive due to energy-saving requirements or other operational constraints. The machine learning model is incrementally updated based on the performance of previous block producers, adjusting feature weights to reduce the likelihood of selecting unreliable nodes. Supervisors then use the updated model to select the next group of block producers, considering active nodes and their current feature profiles. This cycle repeats continuously, allowing the model to adapt incrementally and become more robust while remaining responsive to dynamic application requirements.

## Periodic group selection

In many conventional consensus mechanisms, a single block producer is selected for each block, and the process repeats for subsequent blocks. Mechanisms that allow multiple block producers are less common and achieving both security and energy efficiency in such cases can be challenging. Ranking-based supervised machine learning models generally evaluate the suitability of all nodes simultaneously and thus enable the selection of multiple block producers at once efficiently. Although the top-ranked node is most suitable, closely ranked nodes are also important and often remain stable over time. Consequently, subsequent rounds may involve similar top candidates, making repeated inferences computationally redundant, particularly in energy-constrained settings such as IoT networks.

Based on these considerations, the proposed PoPI consensus mechanism selects a group of top-ranked nodes as block producers across multiple consecutive rounds. This addresses a key limitation of ML-based ranking methods: computational cost, since scoring typically requires separate calculations for each node in every selection cycle. As networks scale to thousands of nodes, per-round inference becomes inefficient and unsustainable [59]. By selecting multiple top-ranked producers in advance, PoPI eliminates repeated inference, conserving energy and improving efficiency. This also reduces the time between consecutive blocks, improving throughput and scalability. This periodic selection is more secure

with an ML model than with rule-based mechanisms, as ML models can adapt to environmental dynamics and form groups of reliable nodes based on past experience. A top-level illustration of this process is shown in Fig 2.

When selecting a group, choosing an optimal size is crucial to ensure both high performance and security. Although particular use cases may select the ideal group size or even treat it as a feature for adaptive sizing, we propose choosing the number of block producers as the square root of the number of active nodes to balance these factors. That is, the block producer size is given by $BP_{size} = \sqrt{N_{active}} = \sqrt{N_{total} - N_{inactive}}$, where $BP_{size}$ denotes the number of block producers, $N_{active}$ is the number of nodes currently capable of producing blocks, $N_{total}$ is the total number of nodes, and $N_{inactive}$ is the number of inactive nodes. Although all nodes ideally participate in consensus, some may remain inactive due to energy-saving requirements or other operational constraints in challenging environments.

A concern arises if upcoming block producers are known in advance due to a strict serial order, as this exposes them to targeted attacks such as DDoS. To mitigate this, block production within the group should avoid fixed or inferable sequences, and each producer should learn its turn only at the moment of selection. To achieve this, PoPI uses the hash of the last block to generate a random value. Since each block includes the hash of its parent and is broadcast after production, all nodes receive the block only after completion. Using this hash to select a group member ensures that producer identities are revealed only after the previous block is completed, leaving little time for targeted attacks. The block hash is converted into an integer ranging from 1 to the number of remaining producers by converting each character to its ASCII value, summing them, and applying a modulus operation. The calculation of the random value can be formulated as the following function.

$$r = \text{generate\_node\_index}(BP_{rem} , h)$$
$$\Rightarrow \left( \sum_{i=1}^{k} \text{ASCII}(h_i) \right) \bmod |BP_{rem}| + 1$$

(1)

In Equation 1, $h_i$ denotes the $i$-th character of the previous block hash, $k$ is the total number of characters in the hash, and $|BP_{rem}|$ represents the number of block producers who have not yet produced a block. The resulting value $r$ corresponds to the selected random node from the remaining block producers.

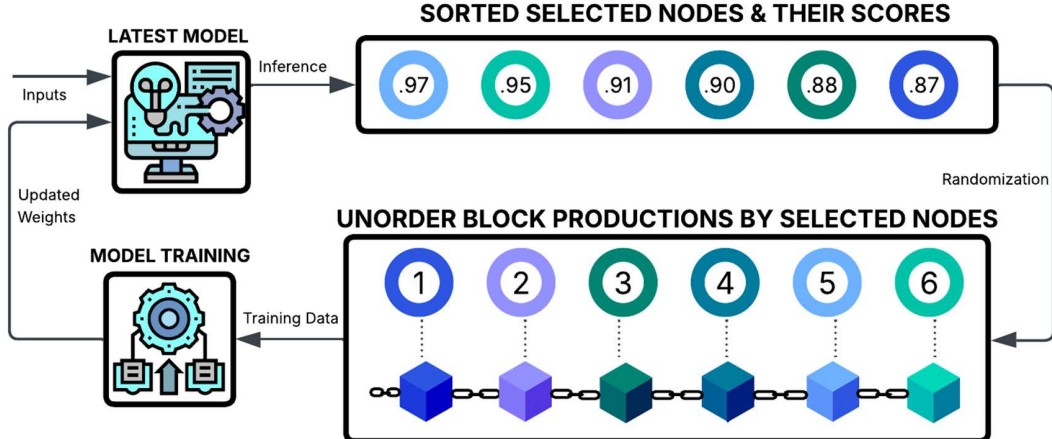

**Fig 2. High-level illustration of how PoPI uses its ML model to rank nodes and select a top-n group from which a randomized method determines their block production order.**

In this context, we can provide some definitions. The machine learning model returns a sorted set of $m$ block producers ($N$), denoted by $BP_{all} = \{N_1, N_2, N_3, \ldots, N_m\}$ and $BP_{size} = |BP_{all}|$. At any given time, if $x$ random nodes ($RN$) have already produced blocks, then $BP_{done} = \{RN_1, RN_2, RN_3, \ldots, RN_x\}$. Consequently, the remaining nodes form the ordered set $BP_{rem} = BP_{all} \setminus BP_{done}$. Each random value $r$ selects the $r$-th element or node from the ordered set $BP_{rem}$. Then, this node is added to the unordered set $BP_{done}$ and $BP_{rem}$ is updated. After each ML inference, the elements of these sets are reset.

## Dynamic feature handling

To assess an IoT device's capability to produce a block, its dynamic attributes, such as battery level, memory usage, and network status, must be considered alongside static features. Since these characteristics vary frequently, they must be evaluated immediately before block producer selection. Relying on a full network-wide broadcast, where every node shares its state with all others, would introduce exponential communication overhead and waste energy. To address this, we introduce a rotating group of supervisors, whose role is analogous to that of election commissioners in traditional voting systems. While the current block producers are active, supervisors gather the latest features and interests of other active nodes. For simplicity, this message, which includes current state data, is referred to as an interest message. The use of these dynamic states is made feasible by the periodic nature of PoPI, as sharing interest messages for each individual block producer would otherwise increase the system overhead and latency.

At any time, a group of block producers operates in a randomized order to generate blocks, while a set of supervisors, selected alongside the producers, collects interest messages from active nodes for the next round. Nodes currently producing blocks may also express interest, but their selection probability is reduced by fair participation mechanisms. Supervisors gather these messages throughout block production and, once the final producer broadcasts its block, they perform inference using the accumulated data to generate a ranked list of nodes. All nodes receive a copy of the inference result from each supervisor. Ideally there should be no variance but if discrepancies occur, nodes accept the most consistent result across supervisors. Introducing multiple supervisors enhances system robustness, as a small, carefully selected group of highly reliable nodes minimizes the risk of failures or malicious behavior.

Fig 3 illustrates the supervisors' duties as duty cycles. Allowing nodes to send data throughout the block production cycle does not create a significant risk of outdated information because the number of block producers is

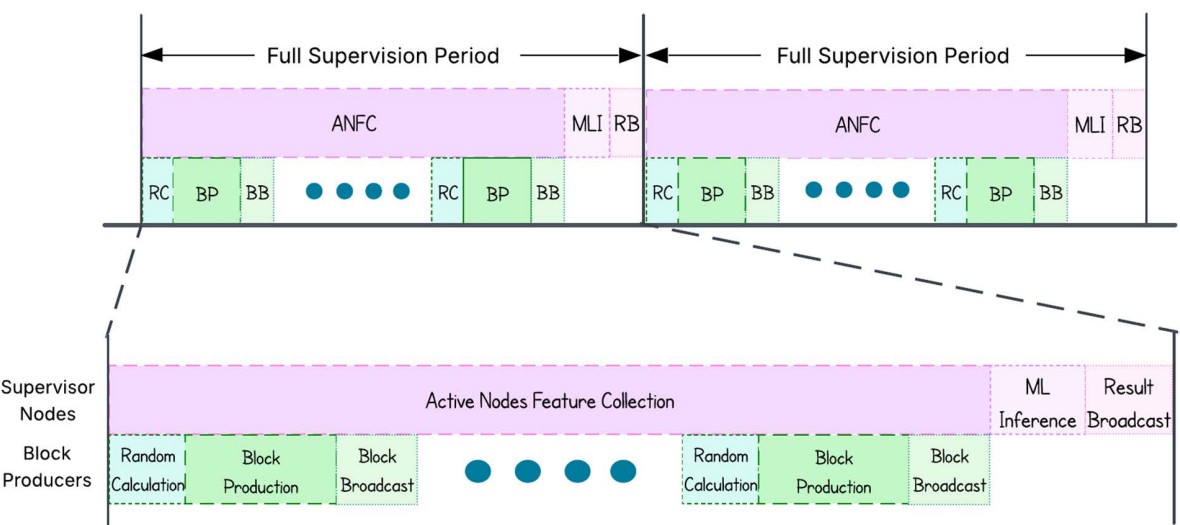

**Fig 3. Illustration of the duties of supervisor nodes during their term.** Supervisors collect node features, run inference after block production, and broadcast results before handing off to the next set of supervisors.

determined using a square root function, which keeps the group size moderate and avoids long delays. Moreover, since the nodes can send the interest messages at any time during the current production cycle, it does not impose a significant short-term load on the network. Short-term changes in node states are handled by the ML model, which continuously adapts to ensure that unreliable nodes are not selected. One important feature of PoPI to ensure security is that all data exchanges are treated as blockchain transactions. Each interest message is digitally signed by the node that creates it. Supervisors also record the ML input used for inference in the blockchain, which is generated by combining the interest messages. These mechanisms allow verification by other nodes and preventing tampering by supervisors.

For proper decentralization, no node should hold a role for an extended period. Hence, supervisors are also required to rotate. Their role lasts only until the current block production group completes, after which they run ML inference to select the next block producers and supervisors. Although stricter criteria or separate mechanisms (e.g., distinct models or interest messages) could be introduced for supervisor selection, our approach requires nodes to have successfully served as block producers at least once to be supervisor. Active nodes submit their interest to current supervisors, who run inference to generate a ranked list. From this list, the top $s$ nodes are selected as supervisors, followed by the next top $n$ nodes as block producers. The number of supervisors follows a base-2 logarithmic relationship with the total number of active nodes, keeping the group size nearly constant and balancing security and communication complexity as the network scales.

Algorithm 1 illustrates the supervisor's role in PoPI. When a node is designated as a supervisor, it continuously collects interest messages from active nodes until the current block production is complete. The collected *data*, representing the status of active nodes, serves as input to the ML model. Based on the inference result $res_{full}$, the top-$s$ nodes meeting the supervisor criteria are selected to form the new supervisor list $sup_{list}$, while the top-$n$ nodes from the remaining set are chosen as block producers $BP_{all}$. Finally, the supervisors broadcast the selected lists and the corresponding data to the network.

**Algorithm 1 Supervisor Role Execution in PoPI**

```
1:  Input: supervisors list sup_list, latest ML model ML_latest
2:  if current node ∈ sup_list then
3:      assume supervisor role
4:  else
5:      return
6:  end if
7:  data ← {}
8:  while last block production not completed do
9:      listen for interest messages
10:     if interest received then
11:         data ← data ∪ signed_interest
12:     end if
13: end while
14: n ← len(data)
15: s ← log₂(n)
16: m ← √n
17: res_full ← ML_latest(data, n)
18: new_sup_list ← res_full.get_top(s, rules = sup_criteria)
19: BP_all ← res_full.get_top(m, exclude = new_sup_list)
20: broadcast(data, new_sup_list, BP_all)
```

## Fair participation assurance

In a decentralized system, fair participation among all honest nodes is essential [60]. Selection should not favor long-standing or previously honest nodes at the expense of newer ones, as capable new participants can strengthen the

blockchain. PoPI, an ML-based mechanism, allows the incorporation of features as needed to ensure fair participation for all nodes. The risk posed by dishonest new nodes is mitigated by the adaptability of the machine learning approach, which can downweight fair participation parameters if many new nodes behave dishonestly. In this subsection, we define three fair participation features: longevity probability, starvation duration and cumulative ML score and suggest additional ones.

**Longevity probability.** A node should be rewarded for remaining in the system for a sufficient duration. Unless the node misbehaves, detected through performance-related features, its probability of selection should increase with its time in the system. However, if this probability grows without limit, new nodes would have little chance of participating. Therefore, the probability must reach a maximum after a certain duration. We refer to this probability as the Longevity Probability (LP), which ensures that new nodes are not unfairly deprioritized once they have been in the system long enough.

Among a few options such as capped linear or exponential saturation, the sigmoid function is preferred for PoPI, as it limits early advantage, increases reward moderately after sustained participation, and maintains proportionality among long-term nodes. The sigmoid function, $f_{sigmoid}(t) = \frac{1}{1+e^{-k(t-t_0)}}$, increases slowly at first, accelerates near the midpoint, and then approaches a maximum value, providing a smooth balance that fairly rewards both new and established nodes. Using this function, the value of LP can be found as follows.

$$LP(n_i) = \frac{1}{1 + e^{-k\left(t_{now} - t_{join(n_i)}\right)}}$$

(2)

In Equation 2, $LP(n_i)$ denotes the longevity probability for node $n_i$. The term $t_{now}$ indicates the current system time, while $t_{join(n_i)}$ refers to the time when node $n_i$ first joined the network. The difference $(t_{now} - t_{join(n_i)})$ therefore measures the total active duration of the node. The growth rate parameter $k$ can be tuned to system requirements or even treated as a learnable feature within the ML model, allowing adaptive adjustment as the system evolves.

**Starvation duration.** The presence of a node over time does not indicate whether it has produced a block recently. The Starvation Duration (SD) feature ensures fairness by gradually increasing the selection probability of nodes that have not produced a block while deprioritizing those that have. This serves two purposes: preventing repeated selection of the same nodes, which leads to centralization, and giving capable nodes deprived of block opportunities a chance to produce blocks. Not all starved nodes are automatically selected, as other features balance poor performance or additional factors.

SD is defined as an integer value that increments by one each time a node expresses interest in block production but is not selected. It resets to zero when the node is chosen. Conversely, if a node remains inactive, the value decreases by one per cycle, down to a minimum of zero, reflecting reduced reliability due to potential energy or connectivity issues. The starvation duration for each node is calculated as follows.

$$SD(n_i) = \begin{cases} SD(n_i) + 1 & \text{if } active \text{ and } not\ selected \\ 0 & \text{if } active \text{ and } is\ selected \\ \max(SD(n_i) - 1, 0) & \text{if } inactive \end{cases}$$

(3)

The starvation duration increases linearly without saturation, unlike the longevity probability, ensuring that the longer a node waits, the higher its probability of being selected. For new nodes, the starvation duration begins at zero upon joining and increases as they express interest, treating them similarly to existing nodes that have just produced a block. If required, an alternative exponential increase could further amplify the selection probability of long-waiting nodes. Note that this feature is not naturally in the range 0–1, which is preferred for supervised models sensitive to feature scales. Hence, these integer features are usually normalized or standardized before being used in such models.

**Cumulative ML score.** Each time an active node expresses an interest in participating in block production, the ML model predicts a suitability score for it, typically in the range 0–1. Instead of discarding these values and letting them go to waste, they are accumulated over multiple rounds to form a cumulative ML score, which serves as a feature reflecting the node's overall suitability without introducing additional overhead. For each node, its current cumulative score is maintained, $ml\text{-}sum_{curr}$, along with a counter $c$ tracking the number of predictions. Upon receiving a new predicted score $s_{latest}$ from the supervisors, the cumulative score is updated by simple addition, and the counter is incremented by one. The update equations are as follows.

$$ml\text{-}sum_{new} = ml\text{-}sum_{curr} + s_{latest},$$
$$c_{new} = c_{curr} + 1 \tag{4}$$

Unlike starvation duration, if a node remains inactive, its cumulative score and counter remain unchanged. When a node is selected as a block producer, these values are reset to 0. Storing the counter $c$ also allows computation of the mean ML score as a separate feature. Furthermore, if individual scores are retained, the variance of ML scores can serve as an additional feature, providing further insight into node behavior over time.

### Consensus as a machine learning model

Up to this point, the machine learning model of PoPI has been presented as a supervised learning framework that integrates multiple features to generate a final suitability score. In this subsection, we examine the internal architecture of the model in greater detail and discuss how the model can be interpreted as a component of the consensus mechanism, effectively framing block producer selection as a machine learning problem.

**Consensus model structure.** In the proposed PoPI consensus mechanism, block producer selection is formulated as a regression-based ranking problem using supervised learning. Each node $j \in \{1, 2, \ldots, N\}$ is represented by a feature vector $x_j \in \mathbb{R}^d$ that captures relevant attributes such as energy state, reliability, hardware configurations, and past participation. The model produces a real-valued prediction $\hat{y}_j = h(x_j; \theta)$, where a higher score indicates greater suitability for block production. The model must learn to optimize its parameters so that the predicted scores closely reflect the true suitability of nodes for block production. Formally, this involves approximating an unknown target function $f : \mathbb{R}^d \to \mathbb{R}$ with a hypothesis function $h(x; \theta)$ parameterized by $\theta$, minimizing a loss function over the training data:

$$\theta^* = \arg\min_{\theta} \sum_{i=1}^{N} \mathcal{L}\left(y_i, h(x_i; \theta)\right) + \Omega(\theta), \tag{5}$$

In Equation 5, $\mathcal{L}$ quantifies the prediction error, while $\Omega(\theta)$ serves as a regularization term to control model complexity and prevent overfitting. Once trained, nodes are ranked according to their predicted scores, and a subset of the top-ranked nodes is selected as block producers. Although $\mathcal{L}$ provides a general measure of prediction error, the choice of regression model significantly affects both predictive performance and computational efficiency. PoPI accommodates a wide range of regression models [61], including linear models such as Linear Regression (LR), Ridge Regression (RR), and LASSO, ensemble-based approaches such as XGBoost and LightGBM and advanced models such as Factorization Machines (FM) and Field-Aware Factorization Machines (FFM). The particular use case can select a suitable model based on robustness and efficiency requirements. In this work, we evaluated PoPI using a Linear Regression model due to its low resource consumption and high scalability with increasing numbers of nodes, as shown in Section Models Resource Consumption.

**Data preparation.** In supervised learning, proper preprocessing of input features is crucial to ensure that the model can learn effectively and that all features contribute appropriately to the output [62]. For the model in PoPI, applying these

preprocessing steps is necessary for several features. For example, signal strength may fall into discrete categories such as low, medium, or high. Such features should be transformed using one-hot encoding. On the other hand, numerical features that are not naturally within the range [0,1] should be standardized or normalized.

Categorical features, which take values from a finite set of discrete categories, are commonly transformed using one-hot encoding. This allows the model to treat each category independently, avoiding an artificial ordinal relationship [63]. For a categorical feature $x_i$ with $K$ categories, one-hot encoding represents it as a $K$-dimensional binary vector $v_i$, where $v_{ij} = 1$ if $x_i$ belongs to category $j$ and $v_{ij} = 0$ otherwise, for $j = 1, \ldots, K$. For numerical features, such as starvation duration for fair participation, standardization or normalization ensures comparable feature scales. Standardization transforms a feature as $x_i^{std} = (x_i - \mu_i)/\sigma_i$, where $\mu_i$ and $\sigma_i$ are the mean and standard deviation of $x_i$. Alternatively, min–max normalization scales a feature to [0,1] as $x_i^{norm} = (x_i - x_i^{min})/(x_i^{max} - x_i^{min})$, where $x_i^{min}$ and $x_i^{max}$ are the minimum and maximum values in the dataset. Both methods prevent features with larger numeric ranges from dominating model training and improve convergence for gradient-based optimization.

**Feature selection.** Feature selection is a critical aspect of the PoPI framework, as it directly influences node reliability assessment and the fairness of block producer selection. Table 1 presents an extended list of potential features organized by category. While the list is comprehensive, individual deployments should analyze feature importance based on application requirements and select a subset of features or add new ones as needed. The model can incorporate multiple categories, capturing performance history, hardware and network availability, configuration reliability, participation fairness, and block-specific information. Each category can contribute uniquely. Performance features can track past block production and enforce penalties for misbehavior. Reliability and availability features can assess current and overall hardware and network conditions. Fair participation features can prevent repetitive selection and enable gradual inclusion of new nodes. Block content features can introduce controlled variability among candidate nodes to enhance unpredictability. Overall, the modular design of PoPI enables it to adapt to diverse IoT deployment scenarios, such as including environment-specific metrics like temperature, humidity, or soil moisture when relevant.

**Initial consensus model generation.** At the onset of an IoT deployment, a reliable machine learning model for node ranking is typically unavailable due to limited training data. To address this, several strategies can be employed to initialize a model capable of making reasonable predictions, which can then be refined as more data becomes available. An approach is to use a pre-trained model from a related domain and fine-tune it for the target application, a process known as transfer learning [64]. This method leverages knowledge from previous applications and is widely used in various domains to reduce the need for extensive training data [65]. Alternatively, a custom model can be generated by simulation, manual annotation of randomly generated datasets, or feature-specific weight assignment [66], providing a customized starting point that aligns with the operational logic and constraints of the intended IoT deployment.

**Table 1. Extended list of potential model features organized by category for PoPI.**

| Category | List of Possible Features |
|---|---|
| Past Performance Features | Successfully Produced Blocks, Average Block Time, Failed Produced Blocks, Reputation Score, General Trust Score, Supervisor Role Count, Model Training Contribution |
| Hardware Availability Features | Remaining Battery Life, Energy Harvesting Rate, Power Charging, Current CPU Usage, Remaining Memory Storage, Free CPU Cores, Power Consumption |
| Network Availability Features | Bandwidth, Throughput, Latency, Packet Loss Rate, Connection Stability, Signal Strength, Link Quality, Network Availability Ratio |
| Configuration Reliability Features | CPU Power, CPU Frequency, Memory Size, Core Count, Cache Size, Battery Power, Charging Capability |
| Fair Participation Features | Starvation Length, Longevity Probability, Mean ML Score, Cumulative ML Score, Selection Frequency, Join Date |
| Block Content Features | Timestamp, Transactions Size, Block Size, Content Hash, Previous Block Hash, Transactions Hash |

In scenarios where no initial model is available or training data is insufficient, a cold startup strategy can be employed, temporarily relying on established consensus mechanisms such as PoW, PoS, or DPoS until sufficient data is collected [58]. Purely rule-based mechanisms, which can incorporate elements of trust, randomness, and repetition penalties, can also serve as an initial solution, balancing fairness, security, and adaptability. However, they cannot function as a long-term solution, as they lack the adaptability to adjust rules and weights over time, unlike an ML-based mechanism. Rather, these approaches ensure that the system can operate effectively from the start while allowing gradual integration of machine-learning-based selection as more operational data becomes available.

**Continuous model training.** Throughout the system lifecycle, the ML model of PoPI adapts to improve the selection of block producers in response to changing network conditions. In each block production cycle, nodes either successfully generate a block or fail, and this outcome serves as the label, while the features used during inference to select a node form the inputs for retraining. In PoPI, supervisors record the node feature information on the blockchain and can be retrieved by anyone and consequently reused for training. Since the training data is recorded on-chain, it benefits from the inherent immutability and transparency of the blockchain, making it resistant to tampering and independently verifiable by participating nodes. This reduces the risk of manipulated training data and improves the overall reliability of the learning process. For efficiency, training is performed after a group of block producers completes their tasks, as initiating training after each individual block provides minimal improvement while consuming additional resources [67].

The training dataset can mathematically be represented by a matrix $\mathcal{D}$ as follows.

$$\mathcal{D} = \begin{bmatrix} x_{11} & x_{12} & \cdots & x_{1d} & y_1 \\ x_{21} & x_{22} & \cdots & x_{2d} & y_2 \\ \vdots & \vdots & \ddots & \vdots & \vdots \\ x_{n1} & x_{n2} & \cdots & x_{nd} & y_n \end{bmatrix}$$

(6)

Here, each row corresponds to a single block producer during a block production cycle. The first $d$ columns of each row, $x_{i1}, x_{i2}, \ldots, x_{id}$, form the feature vector describing the state of node $i$ at the time when it was selected as a block producer. The last column, $y_i$, serves as the label indicating the outcome of the node's block production attempt. A label of 1 is assigned if the node successfully produces a valid block, and a label of 0 is assigned if the node fails or produces an invalid block. Here, this label can be considered as a form of weak label [68] because training labels are automatically derived from observable system outcomes (i.e., block production outcomes) rather than manually verified ground-truth annotations (i.e., network delay, temporary failure, or malicious behavior). However, benign transient failures are unlikely to be permanently penalized by this labeling as the model learns from aggregated historical behavior over multiple rounds rather than isolated events, whereas persistent malicious behavior is reflected through consistent negative outcomes.

**Model management.** PoPI is most suitable for application-oriented IoT use cases. Unlike mechanisms designed primarily for cryptocurrencies, its main objective is to establish trust and ensure the integrity of critical data in domains such as healthcare, agriculture, smart cities, and environmental monitoring. By leveraging Blockchain as a Service (BaaS) platforms, which provide managed infrastructure, development tools, and cloud-based services, such integration can be achieved seamlessly, allowing blockchain functionalities to be incorporated into existing IoT infrastructures without significant overhead [69]. In private deployments, an upper-layer server can coordinate the training process and manage device features. However, it does not store the blockchain data, thereby preserving decentralization. In public systems, training and feature management can instead be carried out using federated learning [70]. Several studies have proposed techniques to enhance the security of federated learning [71], and the proposed mechanism is sufficiently flexible to incorporate these approaches as needed.

## Theoretical analysis

In this section, we first provide a theoretical analysis of the computational improvements of PoPI, and then discuss how it preserves key security aspects of blockchain. Next, we analyze the CAP theorem in the context of PoPI, which is a critical consideration for any distributed data storage system. Finally, we provide a comparative analysis of the design contributions of PoPI with those of other mechanisms.

### Computational complexity analysis

Two primary aspects are considered to analyze the computational complexity of PoPI: block producer selection complexity and communication complexity. The former quantifies the computational effort required to select a block producer for generating one block, while the latter measures the total communication overhead incurred during this process.

In PoPI, a single inference operation of the machine learning model ranks all participating nodes based on their suitability for block production. Let $N$ denote the total number of nodes in the network. The model produces a ranked list of all $N$ nodes in $\mathcal{O}(N)$ time. From this ranked list, the top $\sqrt{N}$ nodes are selected as potential block producers in a single inference step. Because the ranking process inherently orders all nodes, multiple selections can be made efficiently without repeated computations. Each subsequent block production for $\sqrt{N}$ rounds involves minor arithmetic operations to generate random sequence of block production within the group which can be assumed to require constant time, $\mathcal{O}(1)$. Therefore, after one inference operation with time complexity $\mathcal{O}(N)$, a total of $\sqrt{N}$ block producers can be selected. The amortized computational cost per block producer selection can thus be expressed as follows.

$$C_{\text{select}} = \frac{\mathcal{O}(N)}{\sqrt{N}} = \mathcal{O}(\sqrt{N}).$$

(7)

Hence, the block producer selection complexity of PoPI is $\mathcal{O}(\sqrt{N})$, indicating that the computational overhead scales sublinearly with the total number of participating nodes, offering improved scalability compared to many existing consensus mechanisms. For most other consensus algorithms, this complexity is in the order of $\mathcal{O}(N)$, as each block producer selection requires an independent computation. For instance, in PoEM, a separate machine learning inference is performed for every new block producer selection, resulting in a linear computational complexity with respect to the number of nodes.

The communication complexity of communication-based consensus mechanisms such as PBFT is generally $\mathcal{O}(N^2)$, as they require multiple rounds of message exchange among all nodes to achieve consensus. In computation-based consensus mechanisms such as PoW, PoS, or PoET, the communication complexity is typically $\mathcal{O}(N)$, as it mainly involves standard transaction propagation within the blockchain network. For PoPI, the communication complexity similarly remains $\mathcal{O}(N)$. Although PoPI introduces additional message exchanges with supervisors, these communications occur as normal blockchain transactions, and the fixed number of supervisors ensures that the complexity does not grow quadratically.

### Security analysis

Blockchain provides IoT applications with several benefits related to security and data integrity. However, since there is no central authority to monitor activities, the consensus protocol itself may be exposed to various security threats. In this subsection, we discuss the robustness of PoPI against these adversarial behavior and how it mitigates potential security concerns.

**Model unpredictability.** A secure consensus mechanism must ensure unpredictability in block producer selection to avoid vulnerabilities, such as targeted attacks. For instance, in PoW, block generation is probabilistic since miners search for a valid nonce through random trials, making the winner unpredictable. Similarly, PoPI achieves unpredictability through a machine learning model, which, being non-explainable, acts as a black box [72]. Additionally, PoPI uses the previous

block hash to determine the internal order within the group, keeping the next block producer's identity unknown until the last moment.

An adversarial behavior could involve attempting to infer the behavior of the distributed model by running the same publicly available inference procedure to predict group selection and target supervisor nodes, partially reducing the effective opacity of the model. However, the PoPI design inherently prevents this behavior. The model weights are updated at the end of each round, prior to the selection of new producers, based on the success or failure of the previous block producers. Hence, the updated parameters become available only immediately before the subsequent selection, limiting advance predictability. Additionally, even if the model parameters were known or did not change significantly, the input data required for inference are not globally accessible. The dynamic features of the nodes change frequently and are transmitted exclusively to the designated supervisors for that round. This prevents adversaries from reconstructing the exact inference inputs and thus predicting node roles.

**Incentive mechanism.** Incentive mechanisms are fundamental in blockchain systems to regulate entity behavior, promote honest participation, and maintain system security and long-term sustainability without centralized control [73]. In PoPI, malicious nodes are disincentivized from misbehavior through a machine learning-based evaluation mechanism that updates node suitability scores based on observed performance. Nodes must execute their assigned tasks correctly and maintain consistent behavior to remain eligible for future selection. Misconduct, such as failing to perform assigned duties or falsifying state information, results in reduced suitability scores and lower future rewards. Under rational assumptions, honest participation therefore becomes the most beneficial strategy, promoting stable cooperative behavior throughout the network [74].

Two major concerns regarding data authenticity are falsified reporting of its state by active nodes and potential manipulation by supervisors. PoPI mitigates these risks by treating all data exchanges as blockchain transactions and requiring supervisors to record the ML inputs used for inference on-chain. This design enables any node to verify that its submitted data was correctly incorporated into the computation, preventing post-hoc denial by nodes and unauthorized modification by supervisors. Since ML inference and group selection are performed across multiple supervisors rather than a single authority, coordinated manipulation would require compromising a significant fraction of supervisors, making such attacks less feasible under partial adversarial assumptions. Misbehaving supervisors are permanently excluded from future participation, creating strong incentives for honest behavior. For general nodes, strategic data exaggeration ultimately reduces credibility and may lead to failed block production when selected, thereby penalizing dishonest reporting. Collectively, this design not only enhances security by mitigating data forgery of node states but also encourages participants to maintain honest behavior to remain part of the system.

**Attack mitigation.** Blockchain systems are vulnerable to various attacks, including Sybil attacks, collusion attacks, liveness attacks, DDoS attacks, and fork-based attacks [75]. The proposed PoPI framework mitigates these threats through a combination of adaptive node evaluation and dynamic selection mechanisms. Unlike static or stake-based selection approaches, PoPI evaluates nodes using multiple performance, reliability, and availability features, making it difficult for adversaries to consistently gain a selection advantage through identity replication or resource concentration, thereby reducing the effectiveness of Sybil attacks. In addition, the presence of multiple supervisor nodes, which rotates across block production cycles, significantly increases the difficulty of coordinated collusion attacks.

Node selection is continuously updated based on historical behavior, so nodes exhibiting abnormal or malicious activity, such as frequent failures or delayed responses, receive lower scores over time, limiting their participation and mitigating liveness disruptions. The dynamic feature-driven selection process, where node states frequently change, also makes it difficult to predict the next block producer group. Within each group, the probabilistic assignment of block producers, which can only be determined after the previous block is finalized, introduces additional variability. Together, these mechanisms reduce predictability and make targeted attacks, including DDoS attempts and fork attacks, significantly harder.

Overall, while PoPI does not eliminate all attack vectors, its adaptive, learning-based approach enhances resilience by continuously refining node selection in response to observed network behavior.

**Miscellaneous concerns.** A potential security concern of PoPI is whether not retraining the model before each block producer selection weakens system integrity. In PoPI, the consensus model is incrementally updated after the group completes, using recent performance data labeled by whether nodes successfully produced blocks along with their selection features. This is similar to mini-batch stochastic gradient descent, where small updates gradually refine the model. Since parameter changes are minimal between consecutive epochs [76], top-ranked nodes general remain stable, preserving selection reliability and system security while avoiding unnecessary computational overhead. An extremely unlikely situation may occur if all supervisors fail, temporarily halting the system. This can be mitigated through a recovery mechanism using backup supervisors or emergency inference with fixed features, ensuring system continuity until normal processes resume.

## CAP theorem analysis

The CAP theorem is a fundamental concept in distributed systems, stating that any distributed data storage system, such as a blockchain, can guarantee only two of three properties: Consistency (C), Availability (A), and Partition Tolerance (P) [77]. In blockchain, consistency means all nodes maintain the same ledger state simultaneously, availability ensures every request receives a valid response even if some nodes fail, and partition tolerance means the system continues operating despite network partitions. Blockchains are inherently partition-tolerant, operating over decentralized, unreliable networks. Hence, they are classified as CP or AP depending on whether they prioritize consistency or availability. For example, Bitcoin prioritizes availability, achieving consistency only in eventual form, making it an AP system [78].

PoPI deterministically selects a group of block producers by running the same model available across all nodes, which ensures consistency in producer selection. The storage of the model inputs on the blockchain further strengthens this consistency. The group-based design improves availability, as another producer within the group can take over if the current one fails. However, PoPI still prioritizes consistency. Availability may be reduced if all group members fail or if a delay occurs during the transition to the next group selection after the current group completes its work. Therefore, under the CAP theorem, PoPI can be classified as a CP-type system. Nevertheless, PoPI achieves better availability than other CP-type mechanisms, such as PoEM, through its group selection strategy.

## Comparative analysis

In this subsection, we clarify the differences between PoPI and other similar consensus mechanisms in terms of their core design contributions. To illustrate the evolution of consensus approaches, Fig 4 highlights how mechanisms released over the years have progressively added new capabilities to block producer selection. PoW ensures strong security through computationally intensive mining but remains energy-inefficient. PoET reduces computational demand using a randomized wait-time mechanism but scales poorly in larger networks. PoEM employs a machine learning model for intelligent block producer selection but incurs high inference overhead and does not adapt to dynamically changing nodes. In contrast, DP-POEM enhances adaptability in dynamic IoT environments by considering dynamic node states through supervisors and achieves improved scalability by leveraging a periodic selection process.

Table 2 provides a comparative analysis of PoPI with existing ML-based consensus mechanisms: PoDL [56], TDCB-D3P [57], CE-PBFT [53], and PoEM [58]. PoDL leverages the computational capabilities of nodes through deep learning to assess reliability but does not account for fairness, allowing high-capacity nodes to overshadow lower-capacity nodes. The other mechanisms evaluate node credibility by considering past performance and using different types of models. Specifically, TDCB-D3P uses Reinforcement Learning (RL), CE-PBFT uses Decision Trees (DT), and both PoEM and PoPI utilize Supervised Learning (SL). However, PoPI is distinct in that it applies the model periodically and incorporates dynamic node features into its selection process. Furthermore, PoPI introduces fairness-aware features that improve

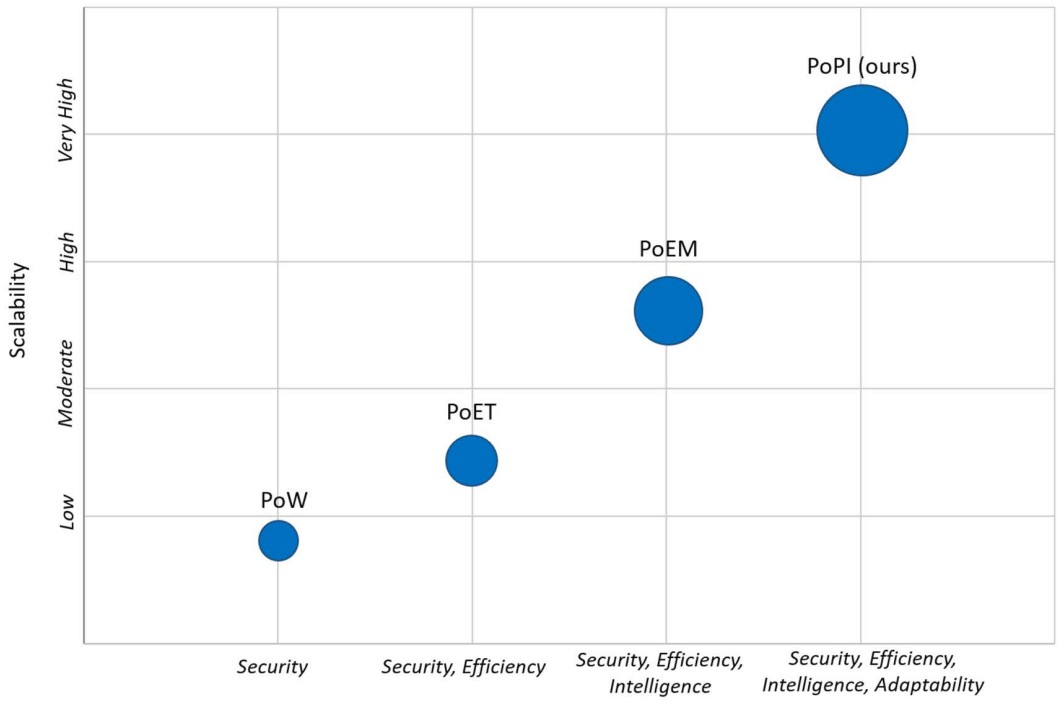

**Fig 4. Evolution of consensus and their capabilities in block producer selection.**

**Table 2. Comparative analysis of PoPI with existing ML-based consensus mechanisms.**

| Consensus | Model Type | Model Purpose | Model Usage | Dynamic Features | Fairness |
|---|---|---|---|---|---|
| PoDL [56] | Deep Learning | Proof of Reliability | Regular | × | Low |
| TDCB-D3P [57] | Reinforcement Learning | Select Delegates | Regular | × | Medium |
| CE-PBFT [53] | Decision Tree | Node Classification | Regular | × | Medium |
| PoEM [58] | Supervised Learning | Select a Producer | Regular | × | Medium |
| **PoPI (ours)** | **Supervised Learning** | **Select Producers** | **Periodic** | ✓ | **High** |

equitable participation, going beyond mere past performance to prevent repeated selection of the same nodes or depriving new nodes of opportunities. Overall, these enhancements make PoPI more adaptable, efficient, and fair in selecting block producers compared to existing ML-based approaches.

## Experimental design

This section outlines the overall experimental design used to evaluate the performance of PoPI, including the simulation settings and performance metrics. We also describe the consensus mechanisms used for comparison and how they were modeled.

### Simulation settings

To evaluate the performance of PoPI, a discrete-event blockchain simulator is developed to emulate a realistic IoT environment. The simulator is implemented in Python 3.11 and executed on a system with an Intel Core i7-1165G7 @ 2.80

GHz processor, 16 GB RAM, and Windows 10. It operates at the transaction level, where transactions are generated at each time unit following a Poisson process. Each node represents an IoT device with dynamic parameters such as battery level, charging and discharging rates, network connectivity, and other operational metrics, as well as fixed attributes including battery capacity, processing power, and storage capacity that influence reliability and consensus participation.

To emulate diverse and dynamic environments, random events such as temporary failure, accelerated battery depletion, halted charging, and reduced connectivity are introduced. These events dynamically affect each node's ability to produce or validate blocks, thereby capturing the instability, heterogeneity, and volatility typical of real-world IoT deployments. Transaction generation, event scheduling, and environmental conditions are kept identical across all consensus mechanisms within a simulation run to ensure fair comparison. Global characteristics such as message propagation delay and block dissemination latency are assumed uniform across nodes. Block production time varies by computational capability, with uniform randomness applied consistently across all mechanisms. Each simulation runs for 10 minutes, with each time unit representing 1 ms. For statistical significance, each configuration is executed 100 times, and results are averaged. Network size is varied to evaluate scalability, using 250, 400, 550, 700, 850, and 1000 nodes. Each block can contain up to 100 transactions.

Evaluating a consensus mechanism in a simulation environment enables controlled, reproducible, and systematic experimentation under varying network and device-level conditions without interference from implementation-specific artifacts. Therefore, full-scale simulation of blockchain systems remains an active research area, with frameworks like BlockSim [79] and JABS [80]. However, these simulators have limited flexibility for novel consensus mechanisms and face compatibility issues with evolving software. They also often lack realistic modeling of IoT nodes under dynamic, resource-constrained conditions, where device failures or intermittent connectivity can impact performance. Therefore, to evaluate PoPI in dynamic IoT settings, we developed a custom simulation environment incorporating stochastic models that approximate real-world device variability and network uncertainty.

## Performance metrics

To evaluate PoPI, four metrics are analyzed: average latency, throughput, computational overhead, and active participation time. For statistical significance, all metrics are measured by averaging their values across all simulation runs.

- **Block Latency:** Block latency is measured as the average time between two successfully produced blocks. This metric reflects the overall efficiency of a blockchain network, as most delays arise from the consensus process. An efficient consensus algorithm is therefore expected to exhibit low latency.

- **Throughput:** Throughput is measured as the total number of transactions successfully recorded per second in the blockchain ledger during the simulation. Excessive time to reach consensus, delays from unreliable block producers, and unconfirmed transactions at the end of the simulation all reduce throughput. Therefore, an efficient consensus mechanism should aim for high throughput.

- **Computational Overhead:** Computational overhead is measured as the average computational effort required by each node during consensus. Most overhead comes from running the consensus protocol, block generation by selected producers, communication, and wasted computations. In resource-constrained IoT environments, minimizing computational effort is crucial for energy efficiency.

- **Active Participation Time:** Active participation time is the proportion of total simulation time during which nodes engage in consensus operations. Continuous computations increase this time, while idle periods when a node knows it will not produce the next block allow IoT devices to conserve energy or perform other tasks. Efficient IoT consensus mechanisms should minimize active participation time while ensuring reliability and security.

**Compared consensus mechanisms**

To evaluate the performance of PoPI, we compare it with five other mechanisms: DT-DPoS, CE-PBFT, PPoR, PoET, and PoEM. Instead of using traditional approaches found to be unsuitable for IoT in previous studies, we select these state-of-the-art mechanisms for their greater potential in IoT applications. This subsection describes each mechanism, including PoPI, and how they were modeled within the simulation framework.

- **DT-DPoS:** Dynamic Trust Delegated Proof of Stake (DT-DPoS) [44] combines DPoS with an EigenTrust-inspired model, where nodes assign local trust scores based on transaction outcomes. These scores are aggregated into global trust values to identify and select reliable witness nodes for block generation and validation.

- **PoET:** Proof of Elapsed Time (PoET) [47] uses a Trusted Execution Environment, such as Intel SGX, to assign each node a cryptographically signed random waiting time. Nodes remain idle for their assigned duration and then become eligible to generate a block, with waiting time ranges increasing with network size to reduce collisions.

- **CE-PBFT:** Credit Evaluation-based Practical Byzantine Fault Tolerance (CE-PBFT) [53] eliminates direct communication among nodes while retaining the core stages of traditional PBFT. It uses the ID3 decision tree to classify nodes, selecting the most secure and responsive as primaries for block generation, with candidate nodes serving as backups and communicating with the primaries.

- **PPoR:** PBFT-based Proof of Reputation (PPoR) [52] divides the network into clusters using K-means clustering. Each cluster has a head, supported by an assistant for fault tolerance. The cluster head is selected based on reputation to create blocks and add local consensus to the global blockchain, while higher-reputation nodes serve as validators to verify the leader's blocks.

- **PoEM:** Proof of Evolutionary Model (PoEM) [58] uses a distributively maintained supervised learning model to rank nodes based on their attributes and blockchain performance. The highest-ranked node generates a block and distributes it across the network. Since attributes are fixed, all nodes can independently compute identical rankings without extra communication. A Linear Regression model with 16 features, trained on previous simulation data, is used for evaluation.

- **PoPI:** The proposed Proof of Periodic Inference (PoPI) selects block producer and supervisor groups using a supervised learning model based on computed ranks. Block producers generate one block at a time in a randomized sequence, while supervisors collect interest messages from active nodes and use these data to determine the next set of producers and supervisors, broadcasting the selection as a blockchain transaction. For consistency with PoEM, PoPI also uses a Linear Regression model, extended with 16 dynamic features, for a total of 32 features.

## Results & discussion

In this section, we present the results of four performance metrics for all six consensus mechanisms, including PoPI, and explain the reasoning behind the observed results. We also highlight the effectiveness of the fair participation feature in PoPI compared to other approaches. Finally, we summarize the overall effectiveness of PoPI.

### Block latency

Fig. 5 illustrates block latency for PPoR, DT-DPoS, CE-PBFT, PoET, PoEM, and PoPI as the number of network nodes increases. PoPI shows the lowest latency among all mechanisms. At 1000 nodes, its latency is 0.25 s, slightly increasing from 0.11 s at 250 nodes. In contrast, PoEM's latency rises from 0.13 s to 0.37 s over the same range, approximately 18% and 48% higher than PoPI. This difference arises primarily because the model is executed every round, increasing PoEM's latency linearly due to inference overhead, whereas PoPI scales sub-linearly. CE-PBFT, DT-DPoS, and PPoR exhibit even higher latencies of 0.50

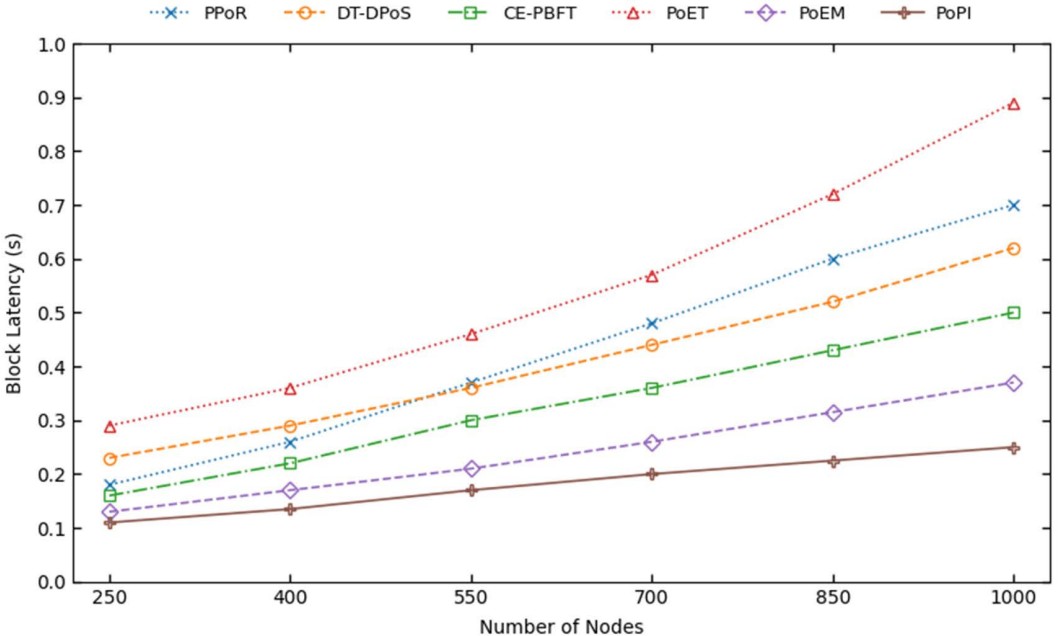

**Fig 5. The block latency of consensus mechanisms with the increase of nodes.**

s, 0.62 s, and 0.71 s, respectively 100%, 144%, and 184% higher than PoPI. This is because they rely on blockchain performance history without considering device-level characteristics. CE-PBFT performs slightly better by reducing communication complexity and more effectively assessing node reliability using decision trees. PoET shows the steepest increase, from 0.28 s at 250 nodes to 0.89 s at 1000 nodes, due to longer randomized wait intervals required to prevent block collisions.

Block latency directly reflects the scalability of a blockchain system, especially in IoT environments with hundreds or thousands of nodes. The lower latency of PoPI demonstrates its improved scalability, mainly because after a single ML inference, it does not perform any computationally expensive operations for an extended period. As nodes gain the right to produce blocks more quickly, they generate blocks with shorter wait times. Consideration of the current states of nodes further minimize delays caused by unreliable nodes, making PoPI highly effective for dynamic IoT environments with low-latency requirements.

In Fig 6, we present the results of 100 independent simulation runs for the 1000-node scenario. The figure shows that the relative performance ordering observed in Fig 5 remains generally stable across runs. In particular, PoPI consistently outperforms PoEM, the second-ranked consensus in terms of mean block latency. This performance advantage can be attributed to the incorporation of dynamic state awareness and supervised learning, which enable more consistent evaluation of node credibility and current system conditions. To further validate the robustness of the results, we report the variance, standard deviation, and 95% confidence intervals for all consensus mechanisms in Table 3. The statistical dispersion of PoPI across 100 runs remains minimal, with a variance of 0.000252, a standard deviation of 0.015865, and a 95% confidence interval of (0.24685, 0.25315). Overall, the results confirm both the consistency of the observed improvements and the stability of the simulation environment.

## Throughput

Fig 7 shows the throughput of all six consensus mechanisms as a function of the number of nodes. All mechanisms exhibit a decrease in throughput as the network grows, since nodes do not generate the transactions themselves in our designed environment.

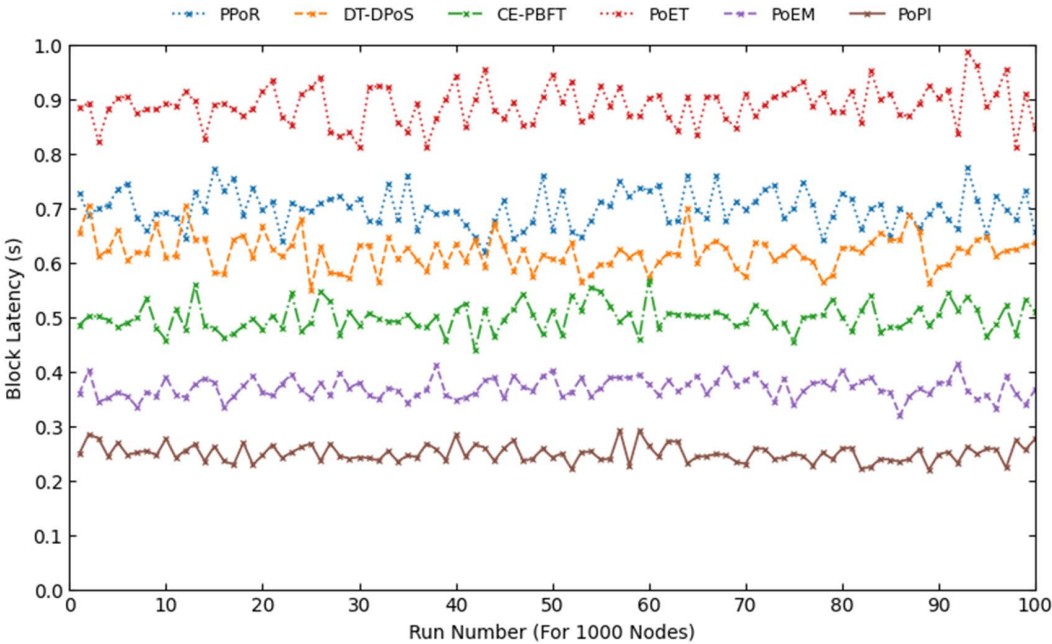

**Fig 6. The block latency of consensus mechanisms in 100 runs for 1000 nodes.**

**Table 3. Statistical summary of block latency at 100 runs for 1000 nodes.**

| Consensus | Variance | Std Dev | 95% Confidence Interval |
|---|---|---|---|
| PPoR | 0.001115 | 0.033399 | (0.69337, 0.70663) |
| DT-DPoS | 0.000990 | 0.031457 | (0.61376, 0.62624) |
| CE-PBFT | 0.000643 | 0.025354 | (0.49497, 0.50503) |
| PoET | 0.001228 | 0.035044 | (0.88305, 0.89695) |
| PoEM | 0.000360 | 0.018963 | (0.36624, 0.37376) |
| PoPI | 0.000252 | 0.015865 | (0.24685, 0.25315) |

The throughput of PoPI decreases from 885 transactions per second (tps) at 250 nodes to 610 tps at 1000 nodes (31% reduction). PoEM starts at 850 tps but drops to 465 tps at 1000 nodes (45% reduction), resulting in PoPI achieving 33% higher throughput than PoEM at large scale. The reason is similar to the observed differences in latency. The group-based mechanism of PoPI allows group members to confirm transactions without selection delay. In addition, transaction delays related to device failures are also significantly reduced due to the dynamic assessment of the current state of IoT devices. All other consensus mechanisms also exhibit significantly lower throughput at 1000 nodes. Specifically, CE-PBFT, DT-DPoS, and PPoR achieve 364 tps, 205 tps, and 174 tps, respectively. The lowest throughput is observed in PoET, which can attributed to the long wait times associated with a high number of nodes. As a result, PoET achieves a throughput of only 85 tps at 1000 nodes.

Throughput provides an alternative perspective on blockchain scalability. Because each block can include only a limited number of transactions, many pending transactions may remain unconfirmed. When blocks are added more frequently, the overall throughput of the system increases. The results show that PoPI achieves the highest throughput by validating the largest number of transactions, making it particularly suitable for environments with high transaction rates.

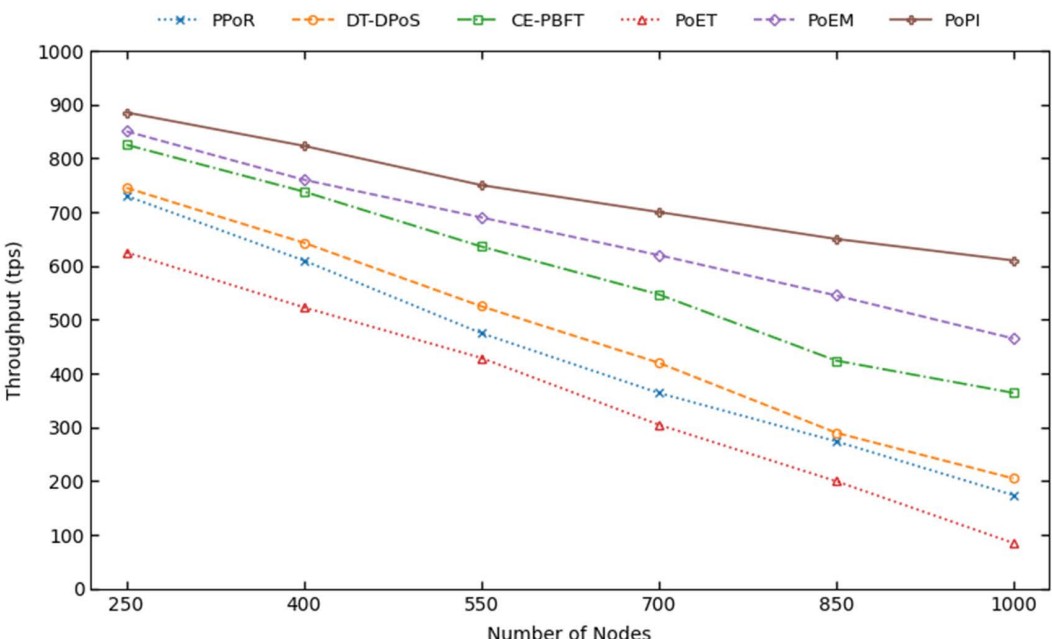

**Fig 7. The throughput of consensus mechanisms with the increase of nodes.**

## Computational overhead

Fig 8 shows the computational overhead of all six consensus mechanisms as the number of nodes increases.

PoPI outperforms all others mechanism in this metric as well. At 1000 nodes, PoPI has an overhead of 149567, increasing only slightly from 124132 at 250 nodes. Although PoET performs poorly in throughput and latency, it achieves a relatively low overhead of 171567 because generating wait times is computationally simpler than supervised learning. However, PoPI still slightly outperforms PoET since its model is used periodically, whereas PoET requires each node to compute a new wait time whenever the previous one finishes, leading to repeated computations. As a result, PoET shows a 32% increase in overhead from 250 to 1000 nodes, compared to 20% for PoPI. PoET may also waste computation due to duplicate wait times and ignoring node reliability. PoEM, which also uses supervised learning, performs inference for every block producer selection, resulting in a higher overhead (204567). The remaining mechanisms exhibit even higher overhead. CE-PBFT and PPoR incur overhead mainly from communication, with CE-PBFT performing better due to reduced messaging and decision tree use. DT-DPoS has the highest overhead and fastest growth rate, as it aggregates trust scores across all node pairs, which leads to quadratic computational complexity.

Resource-constrained IoT devices are already limited in terms of computation. Having lower computational overhead increases the efficiency of a consensus algorithm, making it more suitable for IoT environments. By using a lightweight supervised learning model and applying it infrequently in a periodic manner, PoPI achieves the highest computational efficiency and reduces the overhead.

## Active participation time

Active participation time evaluates the energy efficiency of a consensus mechanism. Fig 9 shows its variation as the number of nodes increases.

Although PoPI outperformed all mechanisms in the other three metrics, PoET achieved a slightly better result at 250 nodes, with an active participation time of 10.5. This occurs because once a node receives its waiting time, it performs no

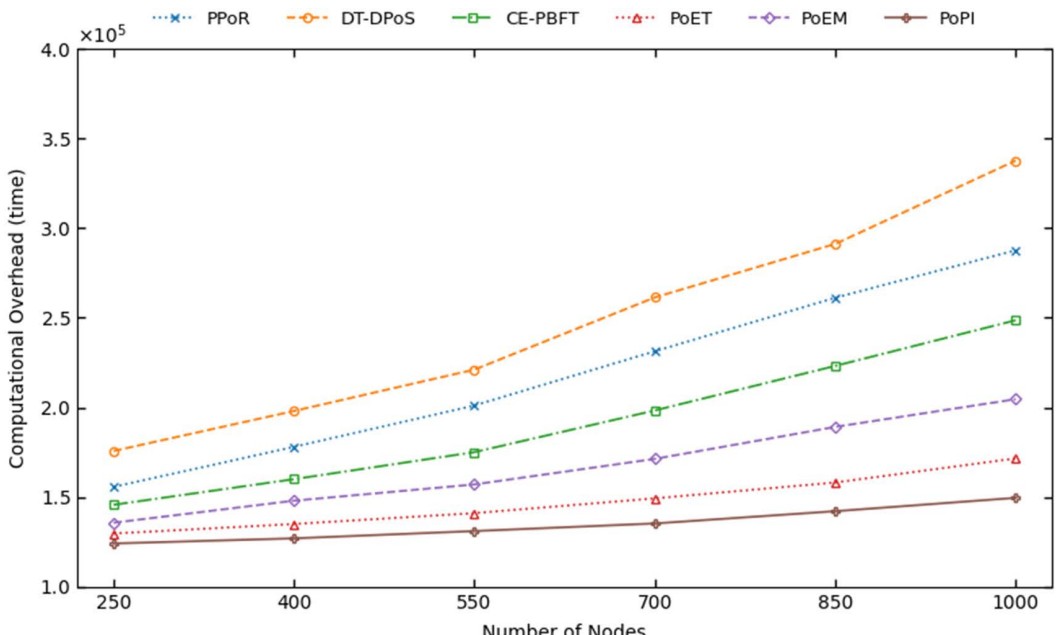

**Fig 8. The computational overhead of consensus mechanisms with the increase of nodes.**

further operations and remains inactive until the period ends. Each node waits, produces a block, receives a new waiting time, and becomes inactive again. This improvement results from trading higher throughput and lower block latency for longer idle periods. PoPI, which achieved the best results in all other metrics, ranked second here, with nodes remaining inactive for about 12.5% of the time at 1000 nodes. The increase in active participation time remains nearly as low as PoET because, although group size grows with the network, the number of inactive nodes grows faster. Group members must stay active in anticipation of their turn, explaining the slight difference from PoET. This does not add extra computational overhead, allowing PoPI to outperform PoET in that aspect. Overall stability in PoPI results from the root-based relation with node count, which prevents linear growth and allows non-group nodes to remain inactive for long periods.

PoEM has an active participation time of 19.5% at 1000 nodes and increases slightly faster than PoPI and PoET. This happens primarily because nodes can only wait until the expected block time of the single top-ranked producer. Although DT-DPoS has the worst computational overhead among the communication-based mechanisms, PPoR and CE-PBFT perform worse in active time because nodes must remain active waiting for messages, which is not included in computational overhead. Overall, DT-DPoS, CE-PBFT, and PPoR have active participation times of 23.9%, 26.6%, and 29.5%, respectively.

Although active participation time does not directly indicate battery, CPU or memory usage of a consensus mechanism, it is a good indication of how much time a node requires to stay active doing high-power-based work and hence deplete its battery. Overall, although PoPI falls slightly behind PoET in active participation time, it still maintains a balanced time with a high number of nodes, making it an energy-efficient consensus mechanism, better than most other optimized solutions.

## Ablation study

PoPI incorporates several novel design choices, each contributing to improvements in multiple aspects of traditional consensus mechanisms. Among them, three core components, namely ML based ranking (MR), periodic grouping (PG), and dynamic feature (DF) utilization, can be attributed to the most significant performance gains. Table 4 presents the ablation

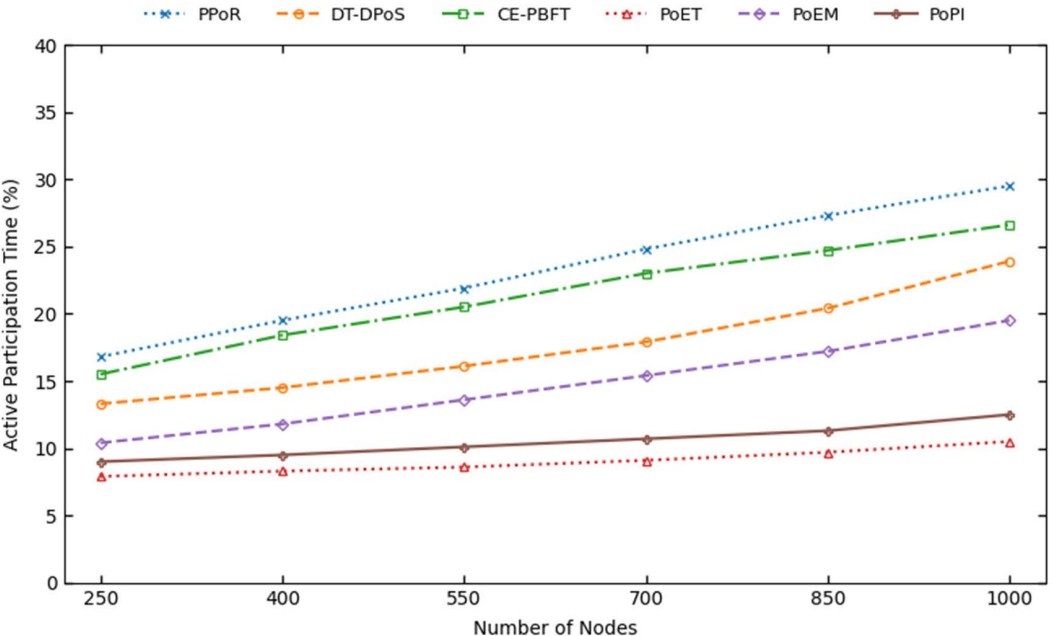

**Fig 9. The active participation time of consensus mechanisms with the increase of nodes.**

**Table 4. Ablation study of different components of PoPI for 1000 nodes.**

| ML Ranking | Periodic Grouping | Dynamic Features | Throughput (tps) |
|---|---|---|---|
| ✗ | ✗ | ✗ | 411 |
| ✗ | ✗ | ✓ | 205 |
| ✗ | ✓ | ✗ | 516 |
| ✗ | ✓ | ✓ | 567 |
| ✓ | ✗ | ✗ | 468 |
| ✓ | ✗ | ✓ | 234 |
| ✓ | ✓ | ✗ | 548 |
| ✓ | ✓ | ✓ | **610** |

analysis of these components in the 1000 nodes scenario. When ML based ranking is disabled, node selection is performed using fixed manually assigned weights for each feature, which remain static over time. Without periodic grouping, a single node is selected in each round. When dynamic features are excluded, the supervisor role is removed, inter node communication is eliminated, and inference is carried out independently by each node using only static information.

The table shows that removing any component leads to a degradation from the original throughput of 610 tps. Among the three components, PG has the most significant impact. When PG is removed while DF is still used, the throughput drops sharply to 234 tps. This occurs because regular DF sharing without periodic coordination incurs substantial communication overhead. When DF is shared periodically, its overall benefits outweigh the associated overhead, indicating that DF is most effective when combined with PG. When both PG and DF are removed but MR is retained, the throughput increases to 468 tps. This occurs because, without DF, no communication is required since the used static features of a node remain unchanged over time. This value is very close to PoEM at 465 tps, as the resulting configuration becomes nearly identical to PoEM.

In comparison, excluding only DF or only MR reduces throughput to 548 tps and 567 tps from 610 tps, respectively. The reduction occurs because, in both cases, the consensus more frequently selects unreliable block producers compared to the complete PoPI framework. This leads to delays in block confirmation and consequently reduces throughput, i.e., the number of transactions included in blocks. Although their immediate impact on throughput appears smaller than PG, their long term importance remains significant. MR enables flexible feature integration and adaptive weight adjustment, improving robustness against adversarial behavior and failures. The absence of only DF leads to a slightly larger performance drop than the absence of MR alone, as ignoring the current node state may result in selecting nodes that are temporarily incapable of generating blocks, thereby causing avoidable delays. In general, the findings validate that PoPI performance improvement is the result of combined and complementary contributions of all design components.

## Fair participation analysis

To demonstrate how the fair participation features of PoPI enhance node participation, we measure how many and how often nodes participate in the blockchain compared to other consensus mechanisms. PoET is excluded because it assigns random wait times to all nodes, ensuring eventual participation and making participation frequency less meaningful. Fig 10 presents two fair participation metrics for a network of 1000 nodes. Node participation percentage represents the proportion of nodes that produced a block at least once. Node participation frequency measures, among nodes that produced a block at least twice, the average number of rounds between consecutive selections, where each round corresponds to the successful production of one block.

In PoPI, 56% of nodes produced a block at least once, which is 21% higher than the second-highest mechanism, PoEM, due to the fair participation feature and group selection process. In mechanisms where only the top-ranked node is selected, some nodes are effectively excluded, as achieving the highest rank is difficult. By selecting a larger group, PoPI allows more nodes to participate, even if they are lower-ranked within the group. Among other mechanisms, trust-based

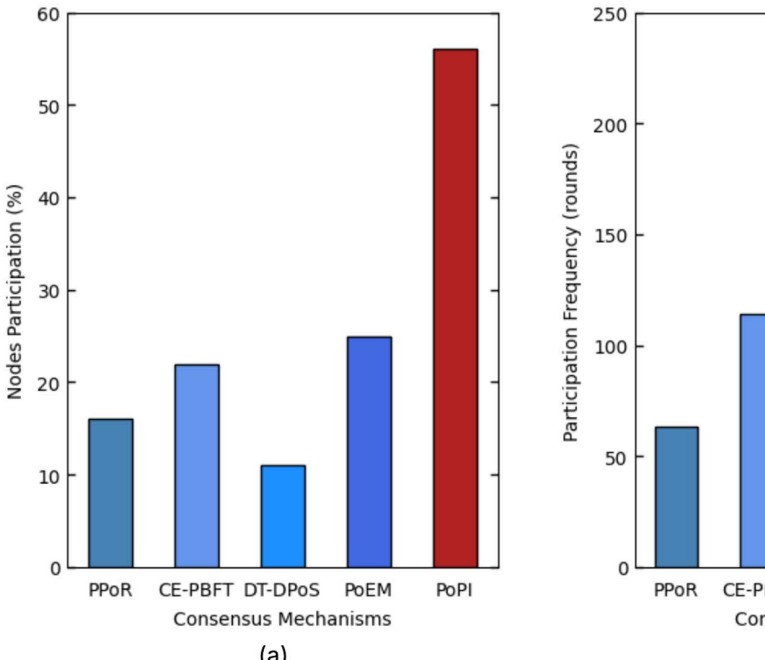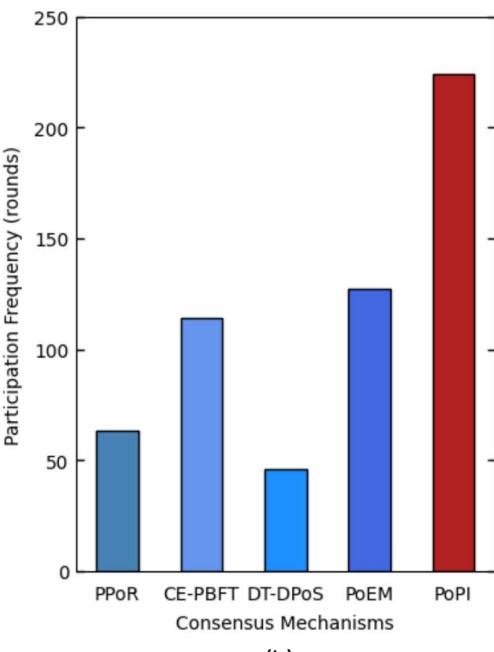

(a)  (b)

**Fig 10. Analysis of fair participation in PoPI for 1000 nodes, showing (a) node participation percentage and (b) participation frequency.**

approaches PPoR and DT-DPoS have lower participation at 16% and 11%, respectively, while CE-PBFT (22%) and PoEM (25%), which use decision trees and supervised learning, achieve slightly higher participation, reflecting increased fairness. PPoR performs better than DT-DPoS due to its clustering mechanism, where nodes in smaller groups have higher participation chances. This pattern is also evident in reselection frequency: for 1000 nodes, a node that produced a block waits 224 confirmed blocks on average before reselection. These results show that PoPI provides significantly fairer participation without compromising security, instead strengthening security and decentralization by involving more reliable nodes.

## Models resource consumption

The results presented for PoPI have so far been evaluated using Linear Regression (LR) as the baseline model. We adopt LR due to its simplicity, interpretability, and low computational overhead. In practice, the choice of the underlying model should depend on application-specific requirements, as different use cases impose varying reliability and efficiency constraints. Since PoPI is designed for deployment in resource-constrained IoT environments, it is important to evaluate the resource consumption characteristics of different models to understand their feasibility in such settings.

To provide a comparative overview, we summarize the model size and latency characteristics of Linear Regression (LR) alongside four additional regression models: Ridge Regression (RR), Gradient Boosting Regressor (GBR), Random Forest Regressor (RFR), and Support Vector Regressor (SVR) in Table 5. The model size reflects the memory footprint of each model, which is crucial for IoT devices with limited memory. CPU time is measured using the *time.process_time()* function in Python and represents the total processor time consumed during model inference. It indicates the computational load imposed on the processor and may exceed the inference time when multiple CPU cores or parallel operations are utilized. Inference time is measured using the *time.time()* function and corresponds to the actual wall-clock time required to execute model inference. In our context, it represents the time required to rank nodes in the blockchain network and directly contributes to the overall block latency. To understand the scalability of each model, we compare these metrics for 400, 700, and 1000 nodes.

Both LR and RR have relatively small model sizes of 1.18 KB and 0.82 KB, respectively. As the node count increases, the model size, CPU time, and inference time remain nearly constant for these models. Similar characteristics are

**Table 5. Resource consumption of different supervised learning models.**

| Model | Node Count | Model Size (KB) | CPU Time (ms) | Inference Time (ms) |
|---|---|---|---|---|
| LR | 400 | 1.18 | 0.6 | 0.3 |
| | 700 | 1.18 | 0.6 | 0.3 |
| | 1000 | 1.18 | 0.7 | 0.4 |
| RR | 400 | 0.82 | 0.5 | 0.3 |
| | 700 | 0.82 | 0.7 | 0.4 |
| | 1000 | 0.82 | 0.7 | 0.4 |
| GBR | 400 | 68.37 | 1.1 | 1.1 |
| | 700 | 69.50 | 1.4 | 1.4 |
| | 1000 | 70.48 | 1.7 | 1.7 |
| RFR | 400 | 1780.38 | 9.1 | 9.1 |
| | 700 | 3113.43 | 12.0 | 12.1 |
| | 1000 | 4445.16 | 15.9 | 16.1 |
| SVR | 400 | 129.29 | 10.7 | 10.1 |
| | 700 | 226.66 | 36.0 | 35.9 |
| | 1000 | 324.43 | 64.1 | 64.4 |

observed for GBR, which also exhibits low CPU and inference time, although with a slightly larger model size of around 70 KB. As shown in Fig 5, PoPI achieves a block latency of 250 ms (0.25 s), implying that the 0.4 ms inference time of LR at 1000 nodes contributes only about 0.16% to the overall latency. In contrast, RFR and SVR require significantly higher computational resources, and their model size, CPU time, and inference time increase approximately linearly with the node count, which may limit scalability. These models typically provide improved modeling capacity when complex feature interactions exist that cannot be captured by simpler models [81]. However, in most blockchain applications that estimate node reliability, such complex relationships are often not required. Therefore, we recommend lightweight and scalable models such as LR, RR, or GBR, provided that their predictive performance satisfies the application requirements.

### Analysis summary

Each consensus mechanism has its own strengths and weaknesses, making it well-suited for certain use cases and less appropriate for others. This diversity reflects the fact that blockchain systems vary significantly in their requirements, including performance, security, scalability, and governance. For highly dynamic IoT environments that require high node reliability, PoPI demonstrates excellent performance across all metrics, highlighting its scalability, applicability, and resource and energy efficiency. Only PoET achieves higher energy efficiency, but performs poorly in terms of scalability and suitability for IoT environments. Therefore, PoET is more appropriate only in scenarios where extreme energy efficiency is required and substantial throughput is not necessary. Some other mechanisms, such as PoEM, do not provide any mechanism to monitor and use dynamic device data. The ablation study shows that adding just dynamic feature sharing to model-based ranking of PoEM is not enough. Rather, it must be complemented by the periodic grouping mechanism of PoPI.

Although most consensus mechanisms face limitations in constantly changing IoT networks, PoPI overcomes these challenges more efficiently and effectively by periodically sharing dynamic features through supervisors. This makes PoPI suitable in a number of IoT applications such as smart cities, environmental monitoring and agriculture. Additionally, the introduction of an optional fair participation mechanism enables a larger number of nodes to contribute to the system without compromising security, and it further enhances decentralization. Therefore, PoPI addresses the issues inherent in centralized IoT architectures and facilitates the integration of blockchain into such systems, making it the most suitable consensus mechanism for IoT applications.

### Conclusion

In this work, we propose PoPI, a lightweight, machine learning-based consensus mechanism that introduces a periodic consensus process through group-based block producer selection and leverages dynamically changing features of IoT devices. This design is particularly suitable for blockchain-enabled IoT platforms, especially those utilizing BaaS frameworks. To efficiently manage dynamic features, we employ a rotating set of supervisor nodes responsible for collecting and sharing device status information. Furthermore, to ensure fair participation across the network, PoPI incorporates mechanisms that prevent repetitive selection of the same nodes while allowing new nodes to gradually join the consensus process. The proposed mechanism is theoretically analyzed in terms of security, consistency, and scalability, and its effectiveness is further validated through extensive numerical simulations. The evaluation results demonstrate that PoPI significantly outperforms existing consensus approaches in scalability, efficiency, and applicability to dynamic IoT environments.

Despite these advances, there exist several limitations in our work. One key limitation is that PoPI has been tested only in a custom simulation environment, which, although allowing controlled experiments, may fail to capture the variability of real-world IoT networks. Therefore, future work can validate the framework using real IoT traces, production-grade deployments, and scenario-specific case studies to further strengthen its practical applicability. Additionally, the supervised model used to select nodes is trained on custom simulation data, which may not fully reflect the suitability of a particular model for analyzing node reliability. Hence, future work may explore alternative machine learning paradigms, such as

unsupervised or reinforcement learning, while maintaining a lightweight design to further enhance the block producer selection process in dynamic IoT environments. Currently, model maintenance and training rely on the capabilities of existing federated learning approaches, which may introduce its own communication overhead or security risks. Hence, another promising direction is the development of efficient methods for distributed training and model generation across decentralized networks, reducing reliance on centralized computation. Beyond these limitations, future studies could also investigate adaptive consensus mechanisms capable of dynamically adjusting parameters or switching strategies based on network conditions, node performance, and device energy levels. Finally, new IoT-oriented consensus designs could incorporate energy optimization techniques directly into the consensus process, achieving higher energy efficiency without compromising security or scalability.

## Author contributions

**Conceptualization:** Mubtasim Kamal Dihan, Abdullah, Amina, Faisal Hussain, Md Sakhawat Hossen.

**Data curation:** Mubtasim Kamal Dihan.

**Formal analysis:** Mubtasim Kamal Dihan, Abdullah, Amina.

**Investigation:** Mubtasim Kamal Dihan, Abdullah, Amina, Md Moniruzzaman, Md Sakhawat Hossen.

**Methodology:** Mubtasim Kamal Dihan, Abdullah, Amina.

**Project administration:** Faisal Hussain, Md Moniruzzaman, Md Sakhawat Hossen.

**Supervision:** Faisal Hussain, Md Moniruzzaman, Md Sakhawat Hossen.

**Validation:** Mubtasim Kamal Dihan, Faisal Hussain.

**Visualization:** Mubtasim Kamal Dihan.

**Writing – original draft:** Mubtasim Kamal Dihan.

**Writing – review & editing:** Mubtasim Kamal Dihan, Faisal Hussain, Md Moniruzzaman, Md Sakhawat Hossen.

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
