## [Decision Letter · Decision Letter 0]

15 Feb 2026

PONE-D-26-01815PoPI: A Machine Learning-Based Consensus Mechanism For Blockchain-Enabled IoT SystemsPLOS One

Dear Dr. Hossen,

Thank you for submitting your manuscript to PLOS ONE. After careful consideration, we feel that it has merit but does not fully meet PLOS ONE’s publication criteria as it currently stands. Therefore, we invite you to submit a revised version of the manuscript that addresses the points raised during the review process.

We look forward to receiving your revised manuscript.

Kind regards,

Yang (Jack) Lu, PhD

Academic Editor

PLOS One

Journal Requirements:

3. We note that your Data Availability Statement is currently as follows: “All relevant data are within the manuscript and its Supporting Information files.”

4. Please remove your figures from within your manuscript file, leaving only the individual TIFF/EPS image files, uploaded separately. These will be automatically included in the reviewers’ PDF.

Reviewers' comments:

Reviewer's Responses to Questions

**Comments to the Author**

1. Is the manuscript technically sound, and do the data support the conclusions?

Reviewer #1: Partly

Reviewer #2: Partly

Reviewer #3: Yes

2. Has the statistical analysis been performed appropriately and rigorously? 

Reviewer #1: Yes

Reviewer #2: Yes

Reviewer #3: Yes

3. Have the authors made all data underlying the findings in their manuscript fully available?

The PLOS Data policy requires authors to make all data underlying the findings described in their manuscript fully available without restriction, with rare exception (please refer to the Data Availability Statement in the manuscript PDF file). The data should be provided as part of the manuscript or its supporting information, or deposited to a public repository. For example, in addition to summary statistics, the data points behind means, medians and variance measures should be available. If there are restrictions on publicly sharing data—e.g. participant privacy or use of data from a third party—those must be specified.requires authors to make all data underlying the findings described in their manuscript fully available without restriction, with rare exception (please refer to the Data Availability Statement in the manuscript PDF file). The data should be provided as part of the manuscript or its supporting information, or deposited to a public repository. For example, in addition to summary statistics, the data points behind means, medians and variance measures should be available. If there are restrictions on publicly sharing data—e.g. participant privacy or use of data from a third party—those must be specified.requires authors to make all data underlying the findings described in their manuscript fully available without restriction, with rare exception (please refer to the Data Availability Statement in the manuscript PDF file). The data should be provided as part of the manuscript or its supporting information, or deposited to a public repository. For example, in addition to summary statistics, the data points behind means, medians and variance measures should be available. If there are restrictions on publicly sharing data—e.g. participant privacy or use of data from a third party—those must be specified.requires authors to make all data underlying the findings described in their manuscript fully available without restriction, with rare exception (please refer to the Data Availability Statement in the manuscript PDF file). The data should be provided as part of the manuscript or its supporting information, or deposited to a public repository. For example, in addition to summary statistics, the data points behind means, medians and variance measures should be available. If there are restrictions on publicly sharing data—e.g. participant privacy or use of data from a third party—those must be specified.

Reviewer #1: Yes

Reviewer #2: Yes

Reviewer #3: Yes

4. Is the manuscript presented in an intelligible fashion and written in standard English?

Reviewer #1: Yes

Reviewer #2: Yes

Reviewer #3: Yes

5. Review Comments to the Author

Reviewer #1: This manuscript proposes PoPI, a machine-learning–based consensus mechanism for blockchain-enabled IoT systems that periodically selects a group of block producers and randomizes block production order within the group. The motivation is clear and relevant, and the overall system architecture is thoughtfully designed. The paper is generally well organized, readable, and technically coherent.

That said, while the approach is promising, there are several issues that currently limit how strongly the experimental results support the authors’ broader claims.

Technical soundness and experimental support.

The idea of amortizing ML inference across multiple blocks and incorporating dynamic device-level features is reasonable, and the simulation results suggest improvements in latency, throughput, and participation efficiency relative to the chosen baselines. However, all evaluations are conducted using a custom discrete-event simulator with modeling assumptions defined by the authors. There is no validation using real IoT traces, real blockchain implementations, or alternative simulation frameworks. As a result, it is difficult to assess how robust the reported gains would be under different or less idealized conditions. Claims regarding scalability and real-world applicability would therefore benefit from more cautious framing or additional supporting evidence.

Statistical rigor.

Although experiments are repeated multiple times and results are averaged, the analysis remains largely descriptive. The manuscript does not report confidence intervals, variance, or statistical significance for the performance comparisons. In addition, there is no sensitivity or ablation analysis to isolate the contribution of individual design choices (e.g., periodic group selection versus the ML-based ranking itself). This makes it difficult to draw strong causal conclusions about which aspects of PoPI are responsible for the observed improvements.

Machine learning component.

While PoPI is positioned as an ML-based consensus mechanism, the evaluation relies solely on a linear regression model with fixed features. There is no comparison across different model classes, no feature importance analysis, and no discussion of issues such as label noise or delayed failure outcomes. Consequently, it remains unclear how much of the performance gain is due to machine learning, as opposed to the group-based scheduling and randomization strategy.

Security considerations.

The security discussion is reasonable at a high level but rests on optimistic assumptions about supervisor honesty and the opacity of the ML model. Potential adversarial behaviors—such as supervisor collusion, strategic misreporting of device features, or partial inference of the model—are not explored in depth. A clearer threat model and discussion of these scenarios would strengthen the contribution.

Overall assessment.

In summary, the manuscript presents a technically plausible and well-motivated approach, but the current experimental and statistical evidence is not sufficient to fully support the breadth of the conclusions. Strengthening the statistical analysis, clarifying the role of the ML component, and better justifying or validating the simulation assumptions would substantially improve the paper.

Reviewer #2: The manuscript proposes a new consensus mechanism called PoPI, which aims to solve the problem of inapplicability of existing consensus mechanisms in the integration of blockchain and Internet of Things under resource-constrained devices and dynamic network conditions. The research questions are of practical significance, however the manuscript still leaves several areas for improvement.

1. It is recommended to clarify more clearly the difference between PoPI and existing similar work. It needs to be clearly stated how PoPI addresses the limitations of mechanisms such as PoEM in dynamic environments.

2. Discussion of related work in the past 1-2 years should be added to the literature review, such as Potential of large language models in blockchain-based supply chain finance [J]. Enterprise Information Systems, 2025: 2541199. and Decentralized finance (DeFi): a paradigm shift in the Fintech [J]. Enterprise Information Systems, 2024, 18(9): 2397630.

3. It is recommended to clearly state the specific supervised learning model used and explain the reasons for selecting this model.

4. The result analysis mostly stays at the data description level, lacking an in-depth explanation of the underlying mechanism.

5. It is recommended to add specific tests for different IoT scenarios to verify the performance of PoPI in different environments.

Reviewer #3: Strengths

- The periodic selection of a group of block producers effectively minimizes the computational overhead typically associated with running machine learning inferences for every single block.

- Incorporating dynamic device characteristics, such as real-time battery levels and network bandwidth, into the consensus selection process represents a highly practical approach to improving IoT network reliability.

- The introduction of fair participation features, such as Longevity Probability and Starvation Duration, ensures a balanced network involvement and mitigates the risk of centralization by preventing the repeated selection of the exact same nodes.

- The experimental setup provides a comprehensive comparison against relevant modern consensus protocols, specifically DT-DPoS, CE-PBFT, PPoR, PoET, and PoEM, across meaningful metrics like throughput, latency, and computational overhead.

Issue:

- The security analysis states that the machine learning model acts as a "black box," making selections unpredictable. However, because the system deterministically selects producers by running the exact same model available across all nodes, an adversary could potentially run the same inference to predict group selection and target supervisor nodes. The authors should address how the protocol prevents this deterministic predictability.

- The mechanism continuously trains the model using block production outcomes as labels, assigning a 1 for success and a 0 for failure. The authors need to clarify how the system handles false negatives or distinguishes between malicious failures and benign failures (e.g., sudden hardware drops), as mislabeling could degrade model accuracy over time.

6. PLOS authors have the option to publish the peer review history of their article (what does this mean?). If published, this will include your full peer review and any attached files.). If published, this will include your full peer review and any attached files.). If published, this will include your full peer review and any attached files.). If published, this will include your full peer review and any attached files.

...

Reviewer #1: No

Reviewer #2: No

Reviewer #3: No

---

## [Author Response · Author response to Decision Letter 1]

5 Mar 2026

Reviewer#1, Concern #1:

This manuscript proposes PoPI, a machine-learning–based consensus mechanism for blockchain-enabled IoT systems that periodically selects a group of block producers and randomizes block production order within the group. The motivation is clear and relevant, and the overall system architecture is thoughtfully designed. The paper is generally well organized, readable, and technically coherent.

That said, while the approach is promising, there are several issues that currently limit how strongly the experimental results support the authors’ broader claims.

Technical soundness and experimental support. The idea of amortizing ML inference across multiple blocks and incorporating dynamic device-level features is reasonable, and the simulation results suggest improvements in latency, throughput, and participation efficiency relative to the chosen baselines. However, all evaluations are conducted using a custom discrete-event simulator with modeling assumptions defined by the authors. There is no validation using real IoT traces, real blockchain implementations, or alternative simulation frameworks. As a result, it is difficult to assess how robust the reported gains would be under different or less idealized conditions. Claims regarding scalability and real-world applicability would therefore benefit from more cautious framing or additional supporting evidence.

Author response:

We thank the reviewer for this insightful comment.

In this study, we adopted a custom discrete-event simulator to ensure controlled experimentation, reproducibility, and systematic evaluation under varying network and device-level conditions. The simulator allows us to isolate the impact of the proposed amortized ML inference and dynamic device-level feature integration without interference from implementation-specific artifacts. We have reported detailed modeling assumptions and parameter settings to enhance transparency and reproducibility.

However, we acknowledge that simulation-based evaluation cannot fully capture all real-world deployment complexities. Although our results demonstrate improvements under diverse simulated conditions, we agree that validation using real IoT traces and production-grade blockchain platforms would provide stronger evidence of robustness of the proposed framework. This remains an important direction for future work that aims to evaluate the feasibility of real-world IoT deployment. We have explicitly stated this as a limitation of our work and clarified the reason behind choosing a discrete-event simulator. In addition, we have revised several parts of the results analysis section to provide a more cautious discussion of real world applicability, taking this limitation into account.

Author action:

Clarified the choice of the discrete-event simulator in Simulation Settings section (Page 25) and added real-world deployment testing as a future direction, along with a discussion of the limitations of the current simulator in the Conclusion Section (Page 39).

Reviewer#1, Concern #2:

Statistical rigor. Although experiments are repeated multiple times and results are averaged, the analysis remains largely descriptive. The manuscript does not report confidence intervals, variance, or statistical significance for the performance comparisons. In addition, there is no sensitivity or ablation analysis to isolate the contribution of individual design choices (e.g., periodic group selection versus the ML-based ranking itself). This makes it difficult to draw strong causal conclusions about which aspects of PoPI are responsible for the observed improvements.

Author response:

Thank you for your thoughtful observations.

As noted by the reviewer, each experiment was conducted over 100 independent runs, and we reported the mean values for all evaluated metrics. We agree that including measures of variability would further strengthen the validity of the results. Therefore, to demonstrate consistency, we have included a new figure (Fig 5) for the first metric, block latency, showing the distribution of results across 100 runs at 1000 nodes. The figure shows that the relative performance ordering remains generally stable across runs. Additionally, we added a new table that reports the variance, standard deviation, and 95% confidence intervals for all consensus mechanisms. The dispersion of PoPI across 100 runs remains minimal, with a variance of 0.000252, a standard deviation of 0.015865, and a 95% confidence interval of (0.24685, 0.25315). Overall, the results confirm both the consistency of the observed improvements and the stability of the simulation environment. Since all run-level data were already available, no additional experiments were required.

To better understand the contribution of individual components of PoPI, we conduct an ablation study by isolating its three key components: ML based ranking (MR), dynamic features (DF), and the periodic grouping (PG) mechanism. The results are presented in a new table (Table 2), and the relevant discussion is added in a new subsection called Ablation Study in the Result Analysis section. A short summary of the discussion is provided here.

The results show that removing any component reduces throughput from the full system value of 610 tps. PG has the largest impact, with throughput dropping to 234 tps when removed while DF is still used, due to communication overhead from frequent DF sharing, highlighting the importance of combining DF with PG. Removing both PG and DF but retaining MR increases throughput to 468 tps because no communication is required. Excluding only DF or only MR reduces throughput to 548 tps and 567 tps from 610 tps, respectively. These findings indicate that while PG drives the largest immediate performance gain, MR and DF provide complementary benefits in several aspects, and that PoPI’s improvement arises from the combined effect of all three components.

Author action:

Added a new figure (Fig 5) and table (Table 1), along with relevant discussion (Page 29 and 30), to provide a statistical summary across different experimental runs. Performed an ablation study in a new table (Table 2) and a new section (Page 34 and 35) to isolate key components and clarify their individual contributions to PoPI.

Reviewer#1, Concern #3:

Machine learning component. While PoPI is positioned as an ML-based consensus mechanism, the evaluation relies solely on a linear regression model with fixed features. There is no comparison across different model classes, no feature importance analysis, and no discussion of issues such as label noise or delayed failure outcomes. Consequently, it remains unclear how much of the performance gain is due to machine learning, as opposed to the group-based scheduling and randomization strategy.

Author response:

Thank you for highlighting this point. We adopt linear regression due to its simplicity, interpretability, and low computational overhead. In practice, the choice of the underlying model should depend on application-specific requirements, as different use cases impose varying reliability and efficiency constraints. However, we acknowledge that since PoPI is designed for deployment in resource-constrained IoT environments, it is important to evaluate the resource consumption characteristics of different models to understand their feasibility in such settings. To provide a comparative overview, we summarize the model size, CPU time and inference time of Linear Regression (LR) alongside four other regression models: Ridge Regression (RR), Gradient Boosting Regressor (GBR), Random Forest Regressor (RFR), and Support Vector Regressor (SVR) in a new table (Table 3). To understand the scalability of each model, we compare these metrics for 400, 700, and 1000 nodes.

The table shows that lightweight models such as LR and RR maintain nearly constant model size, CPU time, and inference time as the node count increases, demonstrating stable and scalable behavior. Gradient Boosting Regressor (GBR) exhibits similar computational efficiency with a slightly larger memory footprint. In contrast, more complex models such as RFR and SVR require significantly higher computational resources, and their resource usage increases approximately linearly with the number of nodes. These models are generally when complex feature interactions exist that cannot be captured by simpler models [1]. Therefore, we recommend lightweight and scalable models such as LR, RR, or GBR, provided that their predictive performance satisfies the application requirements.

Regarding whether the performance gain is attributable to the machine learning component, as discussed in response to Concern 2, the ablation study shows that excluding the machine learning module (MR) reduces throughput from 610 to 567 tps (Table 2). PoPI achieves a block latency of 250 ms (0.25 s) at 1000 nodes, implying that the 0.4 ms inference time of LR at 1000 nodes (as shown in Table 3) contributes only about 0.16% to the overall latency. This demonstrates that while the machine learning component minimally increases block latency, it contributes meaningfully to the observed performance improvement.

Author action:

Added a new table (Table 3) along with a dedicated section (Page 36-38) to discuss the resource consumption of different models and clarify the rationale behind the model choice. Also clarified the performance gains attributable to machine learning by conducting a new ablation study (Page 34-35).

Reviewer#1, Concern #4:

Security considerations. The security discussion is reasonable at a high level but rests on optimistic assumptions about supervisor honesty and the opacity of the ML model. Potential adversarial behaviors, such as supervisor collusion, strategic misreporting of device features, or partial inference of the model, are not explored in depth. A clearer threat model and discussion of these scenarios would strengthen the contribution.

Author response:

Thank you for this comment.

We agree that the security discussion of PoPI benefits from a more detailed analysis of potential adversarial behaviors. To address this concern, we have restructured the security analysis section into multiple dedicated subsections, each explicitly discussing different threat scenarios and how PoPI mitigates them. Additionally, we have added new discussions that address issues such as supervisor collusion, strategic misreporting of device features, and the limited opacity of the ML model, together with the corresponding mitigation strategies of PoPI.

Author action:

Reorganized the security analysis section for better clarity and added new discussions addressing various potential security concerns (Page 22-24).

Reviewer#1, Concern #5:

Overall assessment. In summary, the manuscript presents a technically plausible and well-motivated approach, but the current experimental and statistical evidence is not sufficient to fully support the breadth of the conclusions. Strengthening the statistical analysis, clarifying the role of the ML component, and better justifying or validating the simulation assumptions would substantially improve the paper.

Author response:

We sincerely thank the reviewers for their time and effort in providing valuable suggestions to improve the quality of the paper.

Author action:

We have rigorously gone through the manuscript and tried to improve it.

Reviewer#2, Concern #1:

The manuscript proposes a new consensus mechanism called PoPI, which aims to solve the problem of inapplicability of existing consensus mechanisms in the integration of blockchain and Internet of Things under resource-constrained devices and dynamic network conditions. The research questions are of practical significance, however the manuscript still leaves several areas for improvement.

It is recommended to clarify more clearly the difference between PoPI and existing similar work. It needs to be clearly stated how PoPI addresses the limitations of mechanisms such as PoEM in dynamic environments.

Author response:

Thank you for this suggestion. We acknowledge that the current manuscript does not clearly state how PoPI addresses the limitations of existing mechanisms in dynamic environments.

In the Analysis Summary section, we elaborate on the difference of PoPI and how it improves over these mechanisms, including PoEM and PoET. In dynamic environments specifically, the problem with PoEM is that it does not introduce any particular method to assess the latest state of IoT devices, which can change very frequently. PoEM relies on fixed hardware data monitored at the start of the device joining process. However, simply sharing these data every time is not efficient, as it introduces significant communication overhead and can degrade performance.

To strengthen this claim, we perform a new ablation study where we isolate its three key components: ML-based ranking, dynamic features, and the periodic grouping mechanism, and present the results in a new table (Table 2) with relevant discussion in a subsection called Ablation Study in the Result Analysis section. The results show that if we remove dynamic features and periodic grouping from PoPI, the throughput becomes equal to 468 tps. This value is very close to PoEM at 465 tps, as the resulting configuration becomes nearly identical to PoEM. When we use dynamic features but without the periodic nature of PoPI, the throughput is reduced massively to 234 tps, lower than 465 tps of PoEM. This shows that adding just dynamic feature sharing to PoEM is not enough. Rather, it should be complemented by the periodic nature of PoPI. These observations demonstrate that by periodically sharing dynamic features through supervisors, PoPI effectively addresses the limitations of mechanisms such as PoEM in dynamic environments.

Author action:

Added a new section for Ablation Study (Page 34-35) to highlight how different components of PoPI address the limitations of existing consensus mechanisms such as PoEM and elaborated the discussion in the Analysis Summary (Page 38-39) section to clarify the differences between PoPI and existing related work.

Reviewer#2, Concern #2:

Discussion of related work in the past 1–2 years should be added to the literature review, such as:

(1) Potential of large language models in blockchain-based supply chain finance [J]. Enterprise Information Systems, 2025: 2541199

(2) Decentralized finance (DeFi): a paradigm shift in the Fintech [J]. Enterprise Information Systems, 2024, 18(9): 2397630.

Author response:

We thank the reviewer for this valuable comment.

We acknowledge that we unintentionally overlooked several recent contributions in the broader blockchain domain. Additionally, we recognize that the manuscript would benefit from a dedicated discussion of blockchain applications beyond cryptocurrency. Our current related work focuses primarily on machine learning within different blockchain applications. Since our proposed consensus mechanism is designed for non-cryptocurrency applications, such as healthcare, environmental monitoring, and agriculture, it is important to position the work within this broader context. To address this, we have added a dedicated subsection titled Blockchain Beyond Cryptocurrency in the Related Work section.

The first suggested paper has been incorporated into the existing Machine Learning in Blockchain subsection. The second suggested paper has been included in the newly created subsection, along with several other relevant and recent contributions in non-cryptocurrency blockchain applications.

Author action:

Several new references and a new subsection have been added in the Related Work (Page 4 and 5) Section, alongside the suggested ones, to strengthen the literature review.

Reviewer#2, Concern #3:

It is recommended to clearly state the specific supervised learning model used and explain the reasons for selecting this model.

Author response:

Thank you for highlighting this point. We have mentioned that the use linear regression (LR) models in subsection Compared Consensus Mechanism under Experimental Design Section. We adopt LR due to its simplicity, interpretability, and low computational overhead. In practice, the choice of the u

---

## [Decision Letter · Decision Letter 1]

8 Mar 2026

PONE-D-26-01815R1PoPI: A Machine Learning-Based Consensus Mechanism For Blockchain-Enabled IoT SystemsPLOS One

Dear Dr. Hossen,

Thank you for submitting your manuscript to PLOS ONE. After careful consideration, we feel that it has merit but does not fully meet PLOS ONE’s publication criteria as it currently stands. Therefore, we invite you to submit a revised version of the manuscript that addresses the points raised during the review process.

We look forward to receiving your revised manuscript.

Kind regards,

Yang (Jack) Lu, PhD

Academic Editor

PLOS One

**Journal Requirements:**

Reviewers' comments:

Reviewer's Responses to Questions

**Comments to the Author**

1. If the authors have adequately addressed your comments raised in a previous round of review and you feel that this manuscript is now acceptable for publication, you may indicate that here to bypass the “Comments to the Author” section, enter your conflict of interest statement in the “Confidential to Editor” section, and submit your "Accept" recommendation.

Reviewer #2: (No Response)

Reviewer #4: All comments have been addressed

2. Is the manuscript technically sound, and do the data support the conclusions?

Reviewer #2: (No Response)

Reviewer #4: Yes

3. Has the statistical analysis been performed appropriately and rigorously? 

Reviewer #2: (No Response)

Reviewer #4: Yes

4. Have the authors made all data underlying the findings in their manuscript fully available?

The PLOS Data policy requires authors to make all data underlying the findings described in their manuscript fully available without restriction, with rare exception (please refer to the Data Availability Statement in the manuscript PDF file). The data should be provided as part of the manuscript or its supporting information, or deposited to a public repository. For example, in addition to summary statistics, the data points behind means, medians and variance measures should be available. If there are restrictions on publicly sharing data—e.g. participant privacy or use of data from a third party—those must be specified.requires authors to make all data underlying the findings described in their manuscript fully available without restriction, with rare exception (please refer to the Data Availability Statement in the manuscript PDF file). The data should be provided as part of the manuscript or its supporting information, or deposited to a public repository. For example, in addition to summary statistics, the data points behind means, medians and variance measures should be available. If there are restrictions on publicly sharing data—e.g. participant privacy or use of data from a third party—those must be specified.requires authors to make all data underlying the findings described in their manuscript fully available without restriction, with rare exception (please refer to the Data Availability Statement in the manuscript PDF file). The data should be provided as part of the manuscript or its supporting information, or deposited to a public repository. For example, in addition to summary statistics, the data points behind means, medians and variance measures should be available. If there are restrictions on publicly sharing data—e.g. participant privacy or use of data from a third party—those must be specified.requires authors to make all data underlying the findings described in their manuscript fully available without restriction, with rare exception (please refer to the Data Availability Statement in the manuscript PDF file). The data should be provided as part of the manuscript or its supporting information, or deposited to a public repository. For example, in addition to summary statistics, the data points behind means, medians and variance measures should be available. If there are restrictions on publicly sharing data—e.g. participant privacy or use of data from a third party—those must be specified.

Reviewer #2: (No Response)

Reviewer #4: Yes

5. Is the manuscript presented in an intelligible fashion and written in standard English?

Reviewer #2: (No Response)

Reviewer #4: Yes

6. Review Comments to the Author

Reviewer #2: The manuscript proposes a machine learning-based consensus mechanism called PoPI, aiming to solve the consensus problem under resource-constrained devices and dynamic network conditions in blockchain-IoT systems. The manuscript is somewhat innovative in terms of theoretical framework and experimental design. However, there are still several areas for improvement in the manuscript.

1. There should be a literature review in Related Works.

2. Although the authors have added ablation studies, the result analysis remains at the level of performance indicators and lacks an in-depth explanation of the internal working principles of the mechanism.

3. There should be a discussion of the limitations of the manuscript in the Conclusion.

Reviewer #4: 1. The paper proposes modeling block producer selection as a supervised learning regression problem and provides a basic mathematical form and loss function framework. However, the theoretical justification for model selection, feature importance, and training data sources is insufficient. It lacks a clear explanation of the theoretical basis or performance comparison for the final model selection, and also lacks discussion on feature contribution or model interpretability, which is particularly important in security-sensitive scenarios like blockchain consensus mechanisms. It is recommended to add an explanation of the rationale for selecting the machine learning model, supplement it with feature importance or interpretability analysis, and provide a more rigorous theoretical explanation of the source and reliability of the model training data.

2. The security analysis section of the paper mainly explains the advantages of PoPI, such as unpredictability, incentive mechanisms, and model update mechanisms, from a theoretical perspective. However, the related analysis remains at the conceptual level. For example, the paper lacks systematic analysis or experimental verification of common blockchain attack scenarios. Furthermore, since the consensus mechanism relies on machine learning models for node selection, if attackers influence the model input by forging node state data, it could pose a potential risk to system security; this issue is discussed only briefly in the paper. Recommendation: 1. Add theoretical analysis or simulation experiments for typical attack scenarios to the security analysis section, and further explain PoPI's defense capabilities against these attacks.

2. Although the paper proposes that PoPI predicts node capabilities and periodically selects block producer groups through machine learning, its overall framework still shares some similarities with existing ML-based consensus mechanisms. The current paper's comparison focuses more on performance metrics, while the explanation of the essential innovations at the mechanism level is insufficient. This may make it difficult for readers to clearly understand PoPI's core contributions compared to existing methods. It is recommended to add a clearer comparative analysis table to the relevant work section, highlighting the essential differences between PoPI and existing ML-based consensus algorithms in terms of mechanism design, dynamic feature modeling, and node fairness mechanisms, and further emphasize the theoretical contributions in the discussion section.

7. PLOS authors have the option to publish the peer review history of their article (what does this mean?). If published, this will include your full peer review and any attached files.). If published, this will include your full peer review and any attached files.). If published, this will include your full peer review and any attached files.). If published, this will include your full peer review and any attached files.

...

Reviewer #2: No

Reviewer #4: No

---

## [Author Response · Author response to Decision Letter 2]

18 Mar 2026

Reviewer #2, Concern #1

There should be a literature review in Related Works.

Author response

We thank the reviewer for this insightful suggestion. We agree that a clearer literature review is essential to properly position the proposed work within the existing body of research. We also acknowledge that some parts of the current Related Works section did not sufficiently describe the working principles of certain consensus mechanisms. To address this, we have revised and expanded the section to provide a more structured discussion of both traditional and modern blockchain consensus approaches, and we have added additional analysis highlighting their limitations in IoT environments.

Author action

The Related Works section (Page 6–7) has been revised to strengthen the literature review and explicitly highlight the research gap that motivates the proposed consensus mechanism.

Reviewer #2, Concern #2

Although the authors have added ablation studies, the result analysis remains at the level of performance indicators and lacks an in-depth explanation of the internal working principles of the mechanism.

Author response

We thank the reviewer for this valuable comment. We agree that the results of the ablation study could benefit from a more detailed discussion of the underlying mechanisms responsible for the observed outcomes. Additionally, we have carefully evaluated and revised the text in the Result Analysis section so that for each reported performance level, a corresponding discussion is provided to explain the working principles behind the observed results.

In addition, to help better understand the working principles of PoPI and related mechanisms, we have added a new subsection titled Comparative Analysis in the Theoretical Analysis section. This subsection includes a new figure (Fig 4) illustrating how successive consensus approaches have progressively added capabilities to block producer selection, and a new table (Table 2) providing a detailed comparison of PoPI with existing ML-based consensus mechanisms. While this discussion remains primarily at a theoretical level, it offers additional context that explains the observed results more comprehensively.

Author action

The Ablation Study in the Result Analysis section (Page 38–39) has been revised for in-depth explanation, and a new Comparative Analysis subsection (Page 27–28), along with a figure (Fig 4, Page 28) and a table (Table 2, Page 28), has been added to provide further insight into the mechanism’s working principles compared to other mechanisms.

Reviewer #2, Concern #3

There should be a discussion of the limitations of the manuscript in the Conclusion.

Author response

We thank the reviewer for this valuable suggestion. We agree that explicitly discussing the limitations of the proposed work strengthens the manuscript. To address this, we have added a dedicated discussion in the Conclusion section that highlights key limitations, including the use of a custom simulation environment, the reliance on federated learning for model maintenance and training, and the suitability of the chosen machine learning paradigm for evaluating node reliability. These additions clarify the scope and constraints of the current work while guiding directions for future improvements.

Author action

The Conclusion section (Page 43) has been revised to include a discussion of the limitations of the manuscript and to outline the corresponding future research directions.

Reviewer #4, Concern #1

The paper proposes modeling block producer selection as a supervised learning regression problem and provides a basic mathematical form and loss function framework. However, the theoretical justification for model selection, feature importance, and training data sources is insufficient. It lacks a clear explanation of the theoretical basis or performance comparison for the final model selection, and also lacks discussion on feature contribution or model interpretability, which is particularly important in security-sensitive scenarios like blockchain consensus mechanisms. It is recommended to add an explanation of the rationale for selecting the machine learning model, supplement it with feature importance or interpretability analysis, and provide a more rigorous theoretical explanation of the source and reliability of the model training data.

Author response

We thank the reviewer for this insightful and constructive comment. We agree that further clarification of the rationale behind model selection, feature design, and training data is important, particularly in the context of security-sensitive blockchain consensus mechanisms.

For evaluating PoPI within the simulation framework, we adopt a Linear Regression (LR) model. This choice is motivated by its low computational overhead and strong scalability with increasing numbers of nodes, which are essential properties for resource-constrained IoT environments. We have clarified this rationale in the Consensus Model Structure subsection within the main section describing the proposed consensus mechanism. In addition, we provide experimental support in the Model Resource Consumption section, where results show that LR maintains stable model size and consistent inference time as the number of nodes increases. We note that, in practice, the choice of the underlying model can be adapted based on application-specific requirements, as different use cases may impose varying trade-offs between reliability and efficiency.

Regarding feature design, we agree that providing clearer guidance is beneficial. While feature importance may vary across deployments and is therefore not fixed within the framework, the proposed approach is designed to flexibly accommodate a wide range of reliability and performance-related features. To better support this, we have added a new table (Table 1) that categorizes potential features into performance history, hardware and network availability, configuration reliability, participation fairness, and block-specific information. Each category reflects a distinct aspect of node behavior, and representative features are provided to guide practical implementation. A corresponding discussion has been included in a new subsection titled Feature Selection.

To further clarify the source and reliability of the training data, we have expanded the Continuous Model Training section. In each block production cycle, nodes either successfully generate a block or fail, and this outcome serves as the label, while the features used during inference form the inputs for retraining. Since this information is recorded on-chain by supervisor nodes, as defined in the PoPI design, it is transparent, tamper-resistant, and can be independently verified by all participants. This significantly reduces the risk of manipulated training data and improves the reliability of the learning process.

Author action

The manuscript has been revised to explicitly justify the selection of the Linear Regression model in the Consensus Model Structure subsection (Page 18), introduce a new table (Table 1, Page 20) and a dedicated Feature Selection subsection (Page 19–20), and expand the Continuous Model Training section (Page 21) to clarify the source, construction, and reliability of the training data.

Reviewer #4, Concern #2

The security analysis section of the paper mainly explains the advantages of PoPI from a theoretical perspective but lacks systematic analysis or experimental verification of common blockchain attack scenarios. It also briefly discusses the risk of forged model inputs. The reviewer recommends adding theoretical or simulation-based analysis of typical attacks.

Author response

We thank the reviewer for this valuable suggestion. We agree that a more detailed discussion of PoPI’s resilience to typical blockchain attacks strengthens the manuscript.

In response, we have added a new subsection titled Attack Mitigation in the Security Analysis section. This subsection discusses major attack vectors, including Sybil attacks, collusion attacks, liveness attacks, DDoS attacks, and fork attacks. We explain how PoPI mitigates these threats through adaptive, feature-driven node evaluation and dynamic selection mechanisms. Regarding potential manipulation of model inputs, the discussion has been expanded in the Incentive Mechanism subsection, which clarifies how recording ML inputs on-chain enables data verification and encourages honest reporting of node states.

Author action

A new subsection titled Attack Mitigation (Page 25–26) has been added, and the discussion of model input integrity has been expanded in the Incentive Mechanism subsection (Page 25).

Reviewer #4, Concern #3

The paper shares similarities with existing ML-based consensus mechanisms and lacks a clear explanation of its core innovations at the mechanism level. The reviewer recommends adding a comparative analysis.

Author response

We thank the reviewer for this constructive recommendation. We agree that highlighting PoPI’s core contributions compared to existing ML-based consensus mechanisms improves clarity.

In response, we have added a new subsection titled Comparative Analysis. This includes a figure (Fig 4) illustrating the evolution of consensus mechanisms and a table (Table 2) comparing PoPI with PoDL, TDCB-D3P, CE-PBFT, and PoEM. The comparison highlights differences in model type, purpose, usage pattern, dynamic feature incorporation, and fairness. The results show that PoPI uniquely combines periodic model usage, dynamic feature integration, and fairness-aware selection.

Author action

A Comparative Analysis subsection (Page 27–28), along with Fig 4 and Table 2 (Page 28), has been added to clearly present PoPI’s unique contributions.

References

[1] A. Guru, B. K. Mohanta, H. Mohapatra, F. Al-Turjman, C. Altrjman, and A. Yadav, “A survey on consensus protocols and attacks on blockchain technology,” Applied sciences, vol. 13, no. 4, p. 2604, 2023.

---

## [Decision Letter · Decision Letter 2]

29 Mar 2026

PoPI: A Machine Learning-Based Consensus Mechanism For Blockchain-Enabled IoT Systems

PONE-D-26-01815R2

Dear Dr. Hossen,

We’re pleased to inform you that your manuscript has been judged scientifically suitable for publication and will be formally accepted for publication once it meets all outstanding technical requirements.

Kind regards,

Yang (Jack) Lu, PhD

Academic Editor

PLOS One

Additional Editor Comments (optional):

Reviewers' comments:

Reviewer's Responses to Questions

**Comments to the Author**

1. If the authors have adequately addressed your comments raised in a previous round of review and you feel that this manuscript is now acceptable for publication, you may indicate that here to bypass the “Comments to the Author” section, enter your conflict of interest statement in the “Confidential to Editor” section, and submit your "Accept" recommendation.

Reviewer #2: (No Response)

Reviewer #4: All comments have been addressed

2. Is the manuscript technically sound, and do the data support the conclusions?

Reviewer #2: (No Response)

Reviewer #4: Yes

3. Has the statistical analysis been performed appropriately and rigorously? 

Reviewer #2: (No Response)

Reviewer #4: Yes

4. Have the authors made all data underlying the findings in their manuscript fully available?

The PLOS Data policy requires authors to make all data underlying the findings described in their manuscript fully available without restriction, with rare exception (please refer to the Data Availability Statement in the manuscript PDF file). The data should be provided as part of the manuscript or its supporting information, or deposited to a public repository. For example, in addition to summary statistics, the data points behind means, medians and variance measures should be available. If there are restrictions on publicly sharing data—e.g. participant privacy or use of data from a third party—those must be specified.requires authors to make all data underlying the findings described in their manuscript fully available without restriction, with rare exception (please refer to the Data Availability Statement in the manuscript PDF file). The data should be provided as part of the manuscript or its supporting information, or deposited to a public repository. For example, in addition to summary statistics, the data points behind means, medians and variance measures should be available. If there are restrictions on publicly sharing data—e.g. participant privacy or use of data from a third party—those must be specified.requires authors to make all data underlying the findings described in their manuscript fully available without restriction, with rare exception (please refer to the Data Availability Statement in the manuscript PDF file). The data should be provided as part of the manuscript or its supporting information, or deposited to a public repository. For example, in addition to summary statistics, the data points behind means, medians and variance measures should be available. If there are restrictions on publicly sharing data—e.g. participant privacy or use of data from a third party—those must be specified.requires authors to make all data underlying the findings described in their manuscript fully available without restriction, with rare exception (please refer to the Data Availability Statement in the manuscript PDF file). The data should be provided as part of the manuscript or its supporting information, or deposited to a public repository. For example, in addition to summary statistics, the data points behind means, medians and variance measures should be available. If there are restrictions on publicly sharing data—e.g. participant privacy or use of data from a third party—those must be specified.

Reviewer #2: (No Response)

Reviewer #4: Yes

5. Is the manuscript presented in an intelligible fashion and written in standard English?

Reviewer #2: (No Response)

Reviewer #4: Yes

6. Review Comments to the Author

Reviewer #2: The manuscript proposes a machine learning-based blockchain IoT system consensus mechanism called PoPI, which provides an innovative solution to the problem that existing consensus mechanisms are difficult to adapt to resource-constrained IoT devices and dynamic network conditions. The writing is standardized, the structure is clear, and the technical details are fully described. The author has well addressed the reviewers' comments.

Reviewer #4: The article is authentic, written in fluent language, and has a clear theme. It effectively reflects the results and reflections of the relevant work and is therefore adopted.

7. PLOS authors have the option to publish the peer review history of their article (what does this mean?). If published, this will include your full peer review and any attached files.). If published, this will include your full peer review and any attached files.). If published, this will include your full peer review and any attached files.). If published, this will include your full peer review and any attached files.

...

Reviewer #2: No

Reviewer #4: No

---

## [Editor Report · Acceptance letter]

PONE-D-26-01815R2

PLOS One

Dear Dr. Hossen,

I'm pleased to inform you that your manuscript has been deemed suitable for publication in PLOS One. Congratulations! Your manuscript is now being handed over to our production team.

Kind regards,

on behalf of

Dr. Yang (Jack) Lu

Academic Editor

PLOS One